# Compositional Reinforcement Learning from Logical Specifications

**Kishor Jothimurugan**
University of Pennsylvania

**Suguman Bansal**
University of Pennsylvania

**Osbert Bastani**
University of Pennsylvania

**Rajeev Alur**
University of Pennsylvania

## Abstract

We study the problem of learning control policies for complex tasks given by logical specifications. Recent approaches automatically generate a reward function from a given specification and use a suitable reinforcement learning algorithm to learn a policy that maximizes the expected reward. These approaches, however, scale poorly to complex tasks that require high-level planning. In this work, we develop a compositional learning approach, called DIRL, that interleaves high-level planning and reinforcement learning. First, DIRL encodes the specification as an abstract graph; intuitively, vertices and edges of the graph correspond to regions of the state space and simpler sub-tasks, respectively. Our approach then incorporates reinforcement learning to learn neural network policies for each edge (sub-task) within a Dijkstra-style planning algorithm to compute a high-level plan in the graph. An evaluation of the proposed approach on a set of challenging control benchmarks with continuous state and action spaces demonstrates that it outperforms state-of-the-art baselines.

## 1 Introduction

Reinforcement learning (RL) is a promising approach to automatically learning control policies for continuous control tasks—e.g., for challenging tasks such as walking [11] and grasping [6], control of multi-agent systems [31, 22], and control from visual inputs [28]. A key challenge facing RL is the difficulty in specifying the goal. Typically, RL algorithms require the user to provide a reward function that encodes the desired task. However, for complex, long-horizon tasks, providing a suitable reward function can be a daunting task, requiring the user to manually compose rewards for individual subtasks. Poor reward functions can make it hard for the RL algorithm to achieve the goal; e.g., it can result in reward hacking [3], where the agent learns to optimize rewards without achieving the goal.

Recent work has proposed a number of high-level languages for specifying RL tasks [5, 29, 24, 34, 19]. A key feature of these approaches is that they enable the user to specify tasks *compositionally*—i.e., the user can independently specify a set of short-term subgoals, and then ask the robot to perform a complex task that involves achieving some of these subgoals. Existing approaches for learning from high-level specifications typically generate a reward function, which is then used by an off-the-shelf RL algorithm to learn a policy. Recent works based on Reward Machines [19, 35] have proposed RL algorithms that exploit the structure of the specification to improve learning. However, these algorithms are based on model-free RL at both the high- and low-levels instead of model-based RL. Model-free RL has been shown to outperform model-based approaches on low-level control tasks [10]; however, at the high-level, it is unable to exploit the large amount of available structure. Thus, these approaches scale poorly to long horizon tasks involving complex decision making.

35th Conference on Neural Information Processing Systems (NeurIPS 2021).

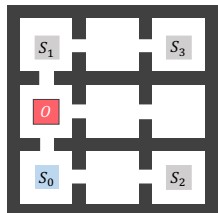 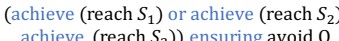

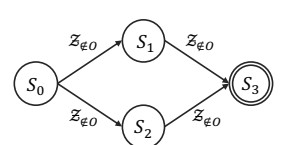 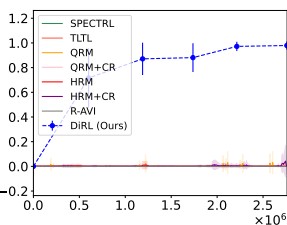

Figure 1: Left: The 9-rooms environment, with initial region $S_0$ in the bottom-left, an obstacle $O$ in the middle-left, and three subgoal regions $S_1, S_2, S_3$ in the remaining corners. Middle top: A user-provided specification $\phi_{\text{ex}}$. Middle bottom: The abstract graph $\mathcal{G}_{\text{ex}}$ DIRL constructs for $\phi_{\text{ex}}$. Right: Learning curves for our approach and some baselines; $x$-axis is number of steps and $y$-axis is probability of achieving $\phi_{\text{ex}}$.

We propose DIRL, a novel compositional RL algorithm that leverages the structure in the specification to decompose the policy synthesis problem into a high-level planning problem and a set of low-level control problems. Then, it interleaves model-based high-level planning with model-free RL to compute a policy that tries to maximize the probability of satisfying the specification. In more detail, our algorithm begins by converting the user-provided specification into an abstract graph whose edges encode the subtasks, and whose vertices encode regions of the state space where each subtask is considered achieved. Then, it uses a Djikstra-style forward graph search algorithm to compute a sequence of subtasks for achieving the specification, aiming to maximize the success probability. Rather than compute a policy to achieve each subtask beforehand, it constructs them on-the-fly for a subtask as soon as Djikstra's algorithm requires the cost of that subtask.

We empirically evaluate[1] our approach on a "rooms environment" (with continuous state and action spaces), where a 2D agent must navigate a set of rooms to achieve the specification, as well as a challenging "fetch environment" where the goal is to use a robot arm to manipulate a block to achieve the specification. We demonstrate that DIRL significantly outperforms state-of-the-art deep RL algorithms for learning policies from specifications, such as SPECTRL, TLTL, QRM and HRM, as well as a state-of-the-art hierarchical RL algorithm, R-AVI, that uses state abstractions, as the complexity of the specification increases. In particular, by exploiting the structure of the specification to decouple high-level planning and low-level control, the sample complexity of DIRL scales roughly linearly in the size of the specification, whereas the baselines quickly degrade in performance. Our results demonstrate that DIRL is capable of learning to perform complex tasks in challenging continuous control environments. In summary, our contributions are as follows:

- A novel compositional algorithm to learn policies for continuous domains from complex high-level specifications that interleaves high-level model-based planning with low-level RL.

- A theoretical analysis of our algorithm showing that it aims to maximize a lower bound on the satisfaction probability of the specification.

- An empirical evaluation demonstrating that our algorithm outperforms several state-of-the-art algorithms for learning from high-level specifications.

**Motivating example.** Consider an RL-agent in the environment of interconnected rooms in Figure 1. The agent is initially in the blue box, and their goal is to navigate to either the top-left room $S_1$ or the bottom-right room $S_2$, followed by the top-right room $S_3$, all the while avoiding the red block $O$. This goal is formally captured by the SPECTRL specification $\phi_{\text{ex}}$ (middle top). This specification is comprised of four simpler RL subtasks—namely, navigating between the corner rooms while avoiding the obstacle. Our approach, DIRL, leverages this structure to improve learning. First, based on the specification alone, it constructs the abstract graph $\mathcal{G}_{\text{ex}}$ (see middle bottom) whose vertices represent the initial region and the three subgoal regions, and the edges correspond to subtasks (labeled with a safety constraint that must be satisfied).

However, $\mathcal{G}_{\text{ex}}$ by itself is insufficient to determine the optimal path—e.g., it does not know that there is no path leading directly from $S_2$ to $S_3$, which is a property of the environment. These differences

---

[1]Our implementation is available at https://github.com/keyshor/dirl.

can be represented as (*a priori* unknown) edge costs in $\mathcal{G}_{\text{ex}}$. At a high level, DIRL trains a policy $\pi_e$ for each edge $e$ in $\mathcal{G}_{\text{ex}}$, and sets the cost of $e$ to be $c(e; \pi_e) = -\log P(e; \pi_e)$, where $P(e; \pi_e)$ is the probability that $\pi_e$ succeeds in achieving $e$. For instance, for the edge $S_0 \to S_1$, $\pi_e$ is trained to reach $S_1$ from a random state in $S_0$ while avoiding $O$. Then, a naïve strategy for identifying the optimal path is to (i) train a policy $\pi_e$ for each edge $e$, (ii) use it to estimate the edge cost $c(e; \pi_e)$, and (iii) run Djikstra's algorithm with these costs.

One challenge is that $\pi_e$ depends on the initial states used in its training—e.g., training $\pi_e$ for $e = S_1 \to S_3$ requires a distribution over $S_1$. Using the wrong distribution can lead to poor performance due to distribution shift; furthermore, training a policy for all edges may unnecessarily waste effort training policies for unimportant edges. To address these challenges, DIRL interweaves training policies with the execution of Djikstra's algorithm, only training $\pi_e$ once Djikstra's algorithm requires the cost of edge $e$. This strategy enables DIRL to scale to complex tasks; in our example, it quickly learns a policy that satisfies the specification with high probability. These design choices are validated empirically—as shown in Figure 1, DIRL quickly learns to achieve the specification, whereas it is beyond the reach of existing approaches.

**Related Work.** There has been recent work on using specifications based on temporal logic for specifying RL tasks [2, 7, 12, 18, 30, 17, 40, 15, 39, 23]. These approaches typically generate a (usually sparse) reward function from a given specification which is then used by an off-the-shelf RL algorithm to learn a policy. In particular, Li et al. [29] propose a variant of Linear Temporal Logic (LTL) called TLTL to specify tasks for robots, and then derive shaped (continuous) rewards from specifications in this language. Jothimurugan et al. [24] propose a specification language called SPECTRL that allows users to encode complex tasks involving sequences, disjunctions, and conjunctions of subtasks, as well as specify safety properties; then, given a specification, they construct a finite state machine called a *task monitor* that is used to obtain shaped (continuous) rewards. Icarte et al. [19] propose an automaton based model called *reward machines* (RM) for high-level task specification and decomposition as well as an RL algorithm (QRM) that exploits this structure. In a later paper [35], they propose variants of QRM including an hierarchical RL algorithm (HRM) to learn policies for tasks specified using RMs. Camacho et al. [9] show that one can generate RMs from temporal specifications but RMs generated this way lead to sparse rewards. Kuo et al. [27] and Vaezipoor et al. [36] propose frameworks for multitask learning using LTL specifications but such approaches require a lot of samples even for relatively simpler environments and tasks. There has also been recent work on using temporal logic specifications for multi-agent RL [16, 33].

More broadly, there has been work on using *policy sketches* [5], which are sequences of subtasks designed to achieve the goal. They show that such approaches can speed up learning for long-horizon tasks. Sun et al. [34] show that providing semantics to the subtasks (e.g., encode rewards that describe when the subtask has been achieved) can further speed up learning. There has also been recent interest in combining high-level planning with reinforcement learning [1, 25, 13]. These approaches all target MDPs with reward functions, whereas we target MDPs with logical task specifications. Furthermore, in our setting, the high-level structure is derived from the given specification, whereas in existing approaches it is manually provided. Illanes et al. [20] propose an RL algorithm for reachability tasks that uses high-level planning to guide low-level RL; however, unlike our approach, they assume that a high-level model is given and high-level planning is not guided by the learned low-level policies. Finally, there has been recent work on applying formal reasoning for extracting interpretable policies [37, 38, 21] as well as for safe reinforcement learning [4, 26].

## 2 Problem Formulation

**MDP.** We consider a *Markov decision process (MDP)* $\mathcal{M} = (S, A, P, \eta)$ with continuous states $S \subseteq \mathbb{R}^n$, continuous actions $A \subseteq \mathbb{R}^m$, transitions $P(s, a, s') = p(s' \mid s, a) \in \mathbb{R}_{\geq 0}$ (i.e., the probability density of transitioning from state $s$ to state $s'$ upon taking action $a$), and initial states $\eta : S \to \mathbb{R}_{\geq 0}$ (i.e., $\eta(s)$ is the probability density of the initial state being $s$). A *trajectory* $\zeta \in \mathcal{Z}$ is either an infinite sequence $\zeta = s_0 \xrightarrow{a_0} s_1 \xrightarrow{a_1} \cdots$ or a finite sequence $\zeta = s_0 \xrightarrow{a_0} \cdots \xrightarrow{a_{t-1}} s_t$ where $s_i \in S$ and $a_i \in A$. A subtrajectory of $\zeta$ is a subsequence $\zeta_{\ell:k} = s_\ell \xrightarrow{a_\ell} \cdots \xrightarrow{a_{k-1}} s_k$. We let $\mathcal{Z}_f$ denote the set of finite trajectories. A (deterministic) *policy* $\pi : \mathcal{Z}_f \to A$ maps a finite trajectory to a fixed action. Given $\pi$, we can sample a trajectory by sampling an initial state $s_0 \sim \eta(\cdot)$, and then iteratively taking the action $a_i = \pi(\zeta_{0:i})$ and sampling a next state $s_{i+1} \sim p(\cdot \mid s_i, a_i)$.

**Specification language.** We consider the specification language SPECTRL for specifying reinforcement learning tasks [24]. A specification $\phi$ in this language is a logical formula over trajectories that indicates whether a given trajectory $\zeta$ successfully accomplishes the desired task. As described below, it can be interpreted as a function $\phi : \mathcal{Z} \to \mathbb{B}$, where $\mathbb{B} = \{\texttt{true}, \texttt{false}\}$.

Formally, a specification is defined over a set of *atomic predicates* $\mathcal{P}_0$, where every $p \in \mathcal{P}_0$ is associated with a function $[\![p]\!] : S \to \mathbb{B}$; we say a state $s$ *satisfies* $p$ (denoted $s \models p$) if and only if $[\![p]\!](s) = \texttt{true}$. For example, given a state $s \in S$, the atomic predicate $[\![\texttt{reach } s]\!](s') = (\|s'-s\| < 1)$ indicates whether the system is in a state close to $s$ with respect to the norm $\|\cdot\|$. The set of *predicates* $\mathcal{P}$ consists of conjunctions and disjunctions of atomic predicates. The syntax of a predicate $b \in \mathcal{P}$ is given by the grammar $b ::= p \mid (b_1 \wedge b_2) \mid (b_1 \vee b_2)$, where $p \in \mathcal{P}_0$. Similar to atomic predicates, each predicate $b \in \mathcal{P}$ corresponds to a function $[\![b]\!] : S \to \mathbb{B}$ defined naturally over Boolean logic. Finally, the syntax of SPECTRL specifications is given by [2]

$$\phi ::= \texttt{achieve } b \mid \phi_1 \texttt{ ensuring } b \mid \phi_1 ; \phi_2 \mid \phi_1 \texttt{ or } \phi_2,$$

where $b \in \mathcal{P}$. In this case, each specification $\phi$ corresponds to a function $[\![\phi]\!] : \mathcal{Z} \to \mathbb{B}$, and we say $\zeta \in \mathcal{Z}$ satisfies $\phi$ (denoted $\zeta \models \phi$) if and only if $[\![\phi]\!](\zeta) = \texttt{true}$. Letting $\zeta$ be a finite trajectory of length $t$, this function is defined by

$$
\begin{aligned}
&\zeta \models \texttt{achieve } b &&\text{if } \exists\, i \le t,\ s_i \models b \\
&\zeta \models \phi \texttt{ ensuring } b &&\text{if } \zeta \models \phi \text{ and } \forall\, i \le t,\ s_i \models b \\
&\zeta \models \phi_1 ; \phi_2 &&\text{if } \exists\, i < t,\ \zeta_{0:i} \models \phi_1 \text{ and } \zeta_{i+1:t} \models \phi_2 \\
&\zeta \models \phi_1 \texttt{ or } \phi_2 &&\text{if } \zeta \models \phi_1 \text{ or } \zeta \models \phi_2.
\end{aligned}
$$

Intuitively, the first clause means that the trajectory should eventually reach a state that satisfies the predicate $b$. The second clause says that the trajectory should satisfy specification $\phi$ while always staying in states that satisfy $b$. The third clause says that the trajectory should sequentially satisfy $\phi_1$ followed by $\phi_2$. The fourth clause means that the trajectory should satisfy either $\phi_1$ or $\phi_2$. An infinite trajectory $\zeta$ satisfies $\phi$ if there is a $t$ such that the prefix $\zeta_{0:t}$ satisfies $\phi$.

We assume that we are able to evaluate $[\![p]\!](s)$ for any atomic predicate $p$ and any state $s$. This is a common assumption in the literature on learning from specifications, and is necessary to interpret a given specification $\phi$.

**Learning from Specifications.** Given an MDP $\mathcal{M}$ with unknown transitions and a specification $\phi$, our goal is to compute a policy $\pi^* : \mathcal{Z}_f \to \mathcal{A}$ such that $\pi^* \in \arg\max_\pi \Pr_{\zeta \sim \mathcal{D}_\pi}[\zeta \models \phi]$, where $\mathcal{D}_\pi$ is the distribution over infinite trajectories generated by $\pi$. In other words, we want to learn a policy $\pi^*$ that maximizes the probability that a generated trajectory $\zeta$ satisfies the specification $\phi$.

We consider the reinforcement learning setting in which we do not know the probabilities $P$ but instead only have access to a simulator of $\mathcal{M}$. Typically, we can only sample trajectories of $\mathcal{M}$ starting at an initial state $s_0 \sim \eta$. Some parts of our algorithm are based on an assumption that we can sample trajectories starting at any state that has been observed before. For example, if taking action $a_0$ in $s_0$ leads to a state $s_1$, we can store $s_1$ and obtain future samples starting at $s_1$.

**Assumption 2.1.** *We can sample $p(\cdot \mid s, a)$ for any previously observed state $s$ and any action $a$.*

# 3 Abstract Reachability

In this section, we describe how to reduce the RL problem for a given MDP $\mathcal{M}$ and specification $\phi$ to a reachability problem on a directed acyclic graph (DAG) $\mathcal{G}_\phi$, augmented with information connecting its edges to subtrajectories in $\mathcal{M}$. In Section 4, we describe how to exploit the compositional structure of $\mathcal{G}_\phi$ to learn efficiently.

## 3.1 Abstract Reachability

We begin by defining the *abstract reachability* problem, and describe how to reduce the problem of learning from a SPECTRL specification to abstract reachability. At a high level, abstract reachability is defined as a graph reachability problem over a directed acyclic graph (DAG) whose vertices

---

[2]Here, $\texttt{achieve}$ and $\texttt{ensuring}$ correspond to the "eventually" and "always" operators in temporal logic.

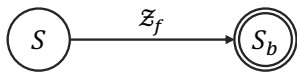

Figure 2: Abstract graph for `achieve` $b$.

correspond to *subgoal regions*—a subgoal region $X \subseteq S$ is a subset of the state space $S$. As discussed below, in our reduction, these subgoal regions are derived from the given specification $\phi$. The constructed graph structure also encodes the relationships between subgoal regions.

**Definition 3.1.** An *abstract graph* $\mathcal{G} = (U, E, u_0, F, \beta, \mathcal{Z}_{\text{safe}})$ is a directed acyclic graph (DAG) with vertices $U$, (directed) edges $E \subseteq U \times U$, initial vertex $u_0 \in U$, final vertices $F \subseteq U$, subgoal region map $\beta : U \to 2^S$ such that for each $u \in U$, $\beta(u)$ is a subgoal region,[3] and *safe trajectories* $\mathcal{Z}_{\text{safe}} = \bigcup_{e \in E} \mathcal{Z}_{\text{safe}}^e$, where $\mathcal{Z}_{\text{safe}}^e \subseteq \mathcal{Z}_f$ denotes the safe trajectories for edge $e \in E$.

Intuitively, $(U, E)$ is a standard DAG, and $u_0$ and $F$ define a graph reachability problem for $(U, E)$. Furthermore, $\beta$ and $\mathcal{Z}_{\text{safe}}$ connect $(U, E)$ back to the original MDP $\mathcal{M}$; in particular, for an edge $e = u \to u'$, $\mathcal{Z}_{\text{safe}}^e$ is the set of trajectories in $\mathcal{M}$ that can be used to transition from $\beta(u)$ to $\beta(u')$.

**Definition 3.2.** An infinite trajectory $\zeta = s_0 \xrightarrow{a_0} s_1 \xrightarrow{a_1} \cdots$ in $\mathcal{M}$ satisfies *abstract reachability* for $\mathcal{G}$ (denoted $\zeta \models \mathcal{G}$) if there is a sequence of indices $0 = i_0 \leq i_1 < \cdots < i_k$ and a path $\rho = u_0 \to u_1 \to \cdots \to u_k$ in $\mathcal{G}$ such that

- $u_k \in F$,
- for all $j \in \{0, \ldots, k\}$, we have $s_{i_j} \in \beta(u_j)$, and
- for all $j < k$, letting $e_j = u_j \to u_{j+1}$, we have $\zeta_{i_j:i_{j+1}} \in \mathcal{Z}_{\text{safe}}^{e_j}$.

The first two conditions state that the trajectory should visit a sequence of subgoal regions corresponding to a path from the initial vertex to some final vertex, and the last condition states that the trajectory should be composed of subtrajectories that are safe according to $\mathcal{Z}_{\text{safe}}$.

**Definition 3.3.** Given MDP $\mathcal{M}$ with unknown transitions and abstract graph $\mathcal{G}$, the *abstract reachability problem* is to compute a policy $\tilde{\pi} : \mathcal{Z}_f \to A$ such that $\tilde{\pi} \in \arg\max_\pi \Pr_{\zeta \sim \mathcal{D}_\pi}[\zeta \models \mathcal{G}]$.

In other words, the goal is to find a policy for which the probability that a generated trajectory satisfies abstract reachability is maximized.

## 3.2 Reduction to Abstract Reachability

Next, we describe how to reduce the RL problem for a given MDP $\mathcal{M}$ and a specification $\phi$ to an abstract reachability problem for $\mathcal{M}$ by constructing an abstract graph $\mathcal{G}_\phi$ inductively from $\phi$. We give a high-level description here, and provide details in Appendix A in the supplement.

First, for each predicate $b$, we define the corresponding subgoal region $S_b = \{s \in S \mid s \models b\}$ denoting the set of states at which $b$ holds. Next, the abstract graph $\mathcal{G}_\phi$ for $\phi = $ `achieve` $b$ is shown in Figure 2. All trajectories in $\mathcal{Z}_f$ are considered safe for the edge $e = u_0 \to u_1$ and the only final vertex is $u_1$ with $\beta(u_1) = S_b$. The abstract graph for a specification of the form $\phi = \phi_1$ `ensuring` $b$ is obtained by taking the graph $\mathcal{G}_{\phi_1}$ and replacing the set of safe trajectories $\mathcal{Z}_{\text{safe}}^e$, for each $e \in E$, with the set $\mathcal{Z}_{\text{safe}}^e \cap \mathcal{Z}_b$, where $\mathcal{Z}_b = \{\zeta \in \mathcal{Z}_f \mid \forall i . s_i \models b\}$ is the set of trajectories in which all states satisfy $b$. For the sequential specification $\phi = \phi_1; \phi_2$, we construct $\mathcal{G}_\phi$ by adding edges from every final vertex of $\mathcal{G}_{\phi_1}$ to every vertex of $\mathcal{G}_{\phi_2}$ that is a neighbor of its initial vertex. Finally, choice $\phi = \phi_1$ `or` $\phi_2$ is handled by merging the initial vertices of the graphs corresponding to the two sub-specifications. Figure 1 shows an example abstract graph. The labels on the vertices are regions in the environment. All trajectories that avoid hitting the obstacle $O$ are safe for all edges. We have the following key guarantee:

**Theorem 3.4.** *Given a* SPECTRL *specification* $\phi$, *we can construct an abstract graph* $\mathcal{G}_\phi$ *such that, for every infinite trajectory* $\zeta \in \mathcal{Z}$, *we have* $\zeta \models \phi$ *if and only if* $\zeta \models \mathcal{G}_\phi$. *Furthermore, the number of vertices in* $\mathcal{G}_\phi$ *is* $O(|\phi|)$ *where* $|\phi|$ *is the size of the specification* $\phi$.

---

[3]We do not require that the subgoal regions partition the state space or that they be non-overlapping.

---
**Algorithm 1** Compositional reinforcement learning algorithm for solving abstract reachability.
---
**function** DIRL($\mathcal{M}, \mathcal{G}$)
    Initialize processed vertices $U_p \leftarrow \varnothing$
    Initialize $\Gamma_{u_0} \leftarrow \{u_0\}$, and $\Gamma_u \leftarrow \varnothing$ for $u \neq u_0$
    Initialize edge policies $\Pi \leftarrow \varnothing$
    **while true do**
        $u \leftarrow$ NEARESTVERTEX$(U \setminus U_p, \Gamma, \Pi)$
        $\rho_u \leftarrow$ SHORTESTPATH$(\Gamma_u)$
        $\eta_u \leftarrow$ REACHDISTRIBUTION$(\rho_u, \Pi)$
        **if** $u \in F$ **then return** PATHPOLICY$(\rho_u, \Pi)$
        **for** $e = u \rightarrow u' \in$ Outgoing$(u)$ **do**
            $\pi_e \leftarrow$ LEARNPOLICY$(e, \eta_u)$
            Add $\rho_u \circ e$ to $\Gamma_{u'}$ and $\pi_e$ to $\Pi$
        Add $u$ to $U_p$
---

We give a proof in Appendix A. As a consequence, we can solve the reinforcement learning problem for $\phi$ by solving the abstract reachability problem for $\mathcal{G}_\phi$. As described below, we leverage the structure of $\mathcal{G}_\phi$ in conjunction with reinforcement learning to do so.

## 4 Compositional Reinforcement Learning

In this section, we propose a compositional approach for learning a policy to solve the abstract reachability problem for MDP $\mathcal{M}$ (with unknown transition probabilities) and abstract graph $\mathcal{G}$.

### 4.1 Overview

At a high level, our algorithm proceeds in three steps:

- For each edge $e = u \rightarrow u'$ in $\mathcal{G}$, use RL to learn a neural network (NN) policy $\pi_e$ to try and transition the system from any state $s \in \beta(u)$ to some state $s' \in \beta(u')$ in a safe way according to $\mathcal{Z}_{\text{safe}}^e$. Importantly, this step requires a distribution $\eta_u$ over initial states $s \in \beta(u)$.

- Use sampling to estimate the probability $P(e; \pi_e, \eta_u)$ that $\pi_e$ safely transitions from $\beta(u)$ to $\beta(u')$.

- Use Djikstra's algorithm in conjunction with the edge costs $c(e) = -\log(P(e; \pi_e, \eta_u))$ to compute a path $\rho^* = u_0 \rightarrow u_1 \rightarrow \cdots \rightarrow u_k$ in $\mathcal{G}$ that minimizes $c(\rho) = -\sum_{j=0}^{k-1} \log(P(e_j; \pi_j, \eta_j))$, where $e_j = u_j \rightarrow u_{j+1}$, $\pi_j = \pi_{e_j}$, and $\eta_j = \eta_{u_j}$.

Then, we could choose $\pi$ to be the sequence of policies $\pi_1, ..., \pi_{k-1}$—i.e., execute each policy $\pi_j$ until it reaches $\beta(u_{j+1})$, and then switch to $\pi_{j+1}$.

There are two challenges that need to be addressed in realizing this approach effectively. First, it is unclear what distribution to use as the initial state distribution $\eta_u$ to train $\pi_e$. Second, it might be unnecessary to learn all the policies since a subset of the edges might be sufficient for the reachability task. Our algorithm (Algorithm 1) addresses these issues by lazily training $\pi_e$—i.e., only training $\pi_e$ when the edge cost $c(e)$ is needed by Djikstra's algorithm.

In more detail, DIRL iteratively processes vertices in $\mathcal{G}$ starting from the initial vertex $u_0$, continuing until it processes a final vertex $u \in F$. It maintains the property that for every $u$ it processes, it has already trained policies for all edges along some path $\rho_u$ from $u_0$ to $u$. This property is satisfied by $u_0$ since there is a path of length zero from $u_0$ to itself. In Algorithm 1, $\Gamma_u$ is the set of all paths from $u_0$ to $u$ discovered so far, $\Gamma = \bigcup_u \Gamma_u$, and $\Pi = \{\pi_e \mid e = u \rightarrow u' \in E, u \in U_p\}$ is the set of all edge policies trained so far. In each iteration, DIRL processes an unprocessed vertex $u$ nearest to $u_0$, which it discovers using NEARESTVERTEX, and performs the following steps:

1. SHORTESTPATH selects the shortest path from $u_0$ to $u$ in $\Gamma_u$, denoted $\rho_u = u_0 \rightarrow \cdots \rightarrow u_k = u$.

2. REACHDISTRIBUTION computes the distribution $\eta_u$ over states in $\beta(u)$ induced by using the sequence of policies $\pi_{e_0}, ..., \pi_{e_{k-1}} \in \Pi$, where $e_j = u_j \to u_{j+1}$ are the edges in $\rho_u$.

3. For every edge $e = u \to u'$, LEARNPOLICY learns a policy $\pi_e$ for $e$ using $\eta_u$ as the initial state distribution, and adds $\pi_e$ to $\Pi$ and $\rho_{u'}$ to $\Gamma_{u'}$, where $\rho_{u'} = u_0 \to \cdots \to u \to u'$; $\pi_e$ is trained to ensure that the resulting trajectories from $\beta(u)$ to $\beta(u')$ are in $\mathcal{Z}_{\text{safe}}^e$ with high probability.

## 4.2 Definitions and Notation

**Edge costs.** We begin by defining the edge costs used in Djikstra's algorithm. Given a policy $\pi_e$ for edge $e = u \to u'$, and an initial state distribution $\eta_u$ over the subgoal region $\beta(u)$, the cost $c(e)$ of $e$ is the negative log probability that $\pi_e$ safely transitions the system from $s_0 \sim \eta_u$ to $\beta(u')$. First, we say a trajectory $\zeta$ starting at $s_0$ *achieves* an $e$ if it safely reaches $\beta(u')$—formally:

**Definition 4.1.** An infinite trajectory $\zeta = s_0 \to s_1 \to \cdots$ *achieves* edge $e = u \to u'$ in $\mathcal{G}$ (denoted $\zeta \models e$) if (i) $s_0 \in \beta(u)$, and (ii) there exists $i$ (constrained to be positive if $u \neq u_0$) such that $s_i \in \beta(u')$ and $\zeta_{0:i} \in \mathcal{Z}_{\text{safe}}^e$; we denote the smallest such $i$ by $i(\zeta, e)$.

Then, the probability that $\pi$ achieves $e$ from an initial state $s_0 \sim \eta_u$ is

$$P(e; \pi_e, \eta_u) = \Pr_{s_0 \sim \eta_u, \zeta \sim \mathcal{D}_{\pi_e, s_0}} [\zeta \models e],$$

where $\mathcal{D}_{\pi_e, s_0}$ is the distribution over infinite trajectories induced by using $\pi_e$ from initial state $s_0$. Finally, the cost of edge $e$ is $c(e) = -\log P(e; \pi_e, \eta_u)$. Note that $c(e)$ is nonnegative for any edge $e$.

**Path policies.** Given edge policies $\Pi$ along with a path $\rho = u_0 \to u_1 \to \cdots \to u_k = u$ in $\mathcal{G}$, we define a *path policy* $\pi_\rho$ to navigate from $\beta(u_0)$ to $\beta(u)$. In particular, $\pi_\rho$ executes $\pi_{u_j \to u_{j+1}}$ (starting from $j = 0$) until reaching $\beta(u_{j+1})$, after which it increments $j \leftarrow j + 1$ (unless $j = k$). That is, $\pi_\rho$ is designed to achieve the sequence of edges in $\rho$. Note that $\pi_\rho$ is stateful since it internally keeps track of the index $j$ of the current policy.

**Induced distribution.** Let path $\rho = u_0 \to \cdots \to u_k = u$ from $u_0$ to $u$ be such that edge policies for all edges along the path have been trained. The induced distribution $\eta_\rho$ is defined inductively on the length of $\rho$. Formally, for the zero length path $\rho = u_0$ (so $u = u_0$), we define $\eta_\rho = \eta$ to be the initial state distribution of the MDP $\mathcal{M}$. Otherwise, we have $\rho = \rho' \circ e$, where $e = u' \to u$. Then, we define $\eta_\rho$ to be the state distribution over $\beta(u)$ induced by using $\pi_e$ from $s_0 \sim \eta_{\rho'}$ conditioned on $\zeta \models e$. Formally, $\eta_\rho$ is the probability distribution over $\beta(u)$ such that for a set of states $S' \subseteq \beta(u)$, the probability of $S'$ according to $\eta_\rho$ is

$$\Pr_{s \sim \eta_\rho} [s \in S'] = \Pr_{s_0 \sim \eta_{\rho'}, \zeta \sim \mathcal{D}_{\pi_e, s_0}} \left[ s_{i(\zeta, e)} \in S' \mid \zeta \models e \right].$$

**Path costs.** The cost of a path $\rho = u_0 \to \cdots \to u_k = u$ is $c(\rho) = -\sum_{j=0}^{k-1} \log P(e_j; \pi_{e_j}, \eta_{\rho_{0:j}})$ where $e_j = u_j \to u_{j+1}$ is the $j$-th edge in $\rho$, and $\rho_{0:j} = u_0 \to \cdots \to u_j$ is the $j$-th prefix of $\rho$.

## 4.3 Algorithm Details

DIRL interleaves Djikstra's algorithm with using RL to train policies $\pi_e$. Note that the edge weights to run Dijkstra's are not given *a priori* since the edge policies and initial state/induced distributions are unknown. Instead, they are computed on-the-fly beginning from the subgoal region $u_0$ using Algorithm 1. We describe each subprocedure below.

**Processing order (NEARESTVERTEX).** On each iteration, DIRL chooses the vertex $u$ to process next to be an unprocessed vertex that has the shortest path from $u_0$—i.e., $u \in \arg\min_{u' \in U \setminus U_p} \min_{\rho \in \Gamma_{u'}} c(\rho)$. This choice is an important part of Djikstra's algorithm. For a graph with fixed costs, it ensures that the computed path $\rho_u$ to each vertex $u$ is minimized. While the costs in our setting are not fixed since they depend on $\eta_u$, this strategy remains an effective heuristic.

**Shortest path computation (SHORTESTPATH).** This subroutine returns a path of minimum cost, $\rho_u \in \arg\min_{\rho \in \Gamma_u} c(\rho)$. These costs can be estimated using Monte Carlo sampling.

**Initial state distribution (REACHDISTRIBUTION).** A key choice DIRL makes is what initial state distribution $\eta_u$ to choose to train policies $\pi_e$ for outgoing edges $e = u \to u'$. DIRL chooses the

initial state distribution $\eta_u = \eta_{\rho_u}$ to be the distribution of states reached by the path policy $\pi_{\rho_u}$ from a random initial state $s_0 \sim \eta$.[4]

**Learning an edge policy (LEARNPOLICY).** Now that the initial state distribution $\eta_u$ is known, we describe how DIRL learns a policy $\pi_e$ for a single edge $e = u \to u'$. At a high level, it trains $\pi_e$ using a standard RL algorithm, where the rewards $\mathbb{1}(\zeta \models e)$ are designed to encourage $\pi_e$ to safely transition the system to a state in $\beta(u')$. To be precise, DIRL uses RL to compute $\pi_e \in \arg\max_\pi P(e; \pi, \eta_u)$. Shaped rewards can be used to improve learning; see Appendix B.

**Constructing a path policy (PATHPOLICY).** Given edge policies $\Pi$ along with a path $\rho = u_0 \to \cdots \to u$, where $u \in F$ is a final vertex, DIRL returns the path policy $\pi_\rho$.

**Theoretical Guarantee.** We have the following guarantee (we give a proof in Appendix C).

**Theorem 4.2.** *Given a path policy $\pi_\rho$ corresponding to a path $\rho = u_0 \to \cdots \to u_k = u$, where $u \in F$, we have* $\Pr_{\zeta \sim \mathcal{D}_{\pi_\rho}}[\zeta \models \mathcal{G}] \geq \exp(-c(\rho))$.

In other words, we guarantee that minimizing the path cost $c(\rho)$ corresponds to maximizing a lower bound on the objective of the abstract reachability problem.

## 5 Experiments

We empirically evaluate our approach on several continuous control environments; details are in Appendix D, E and F.

**Rooms environment.** We consider the 9-Rooms environment shown in Figure 1, and a similar 16-Rooms environment. They have states $(x, y) \in \mathbb{R}^2$ encoding 2D position, actions $(v, \theta) \in \mathbb{R}^2$ encoding speed and direction, and transitions $s' = s + (v\cos(\theta), v\sin(\theta))$. For 9-Rooms, we consider specifications similar to $\phi_{\text{ex}}$ in Figure 1. For 16-Rooms, we consider a series of increasingly challenging specifications $\phi_1, ..., \phi_5$; each $\phi_i$ encodes a sequence of $i$ sub-specifications, each of which has the same form as $\phi_{\text{ex}}$ (see Appendix E). We learn policies using ARS [32] with shaped rewards (see Appendix B); each one is a fully connected NN with 2 hidden layers of 30 neurons each.

**Fetch environment.** We consider the Fetch-Pick-And-Place environment in OpenAI Gym [8], consisting of a robotic arm that can grasp objects and a block to manipulate. The state space is $\mathbb{R}^{25}$, which includes components encoding the gripper position, the (relative) position of the object, and the distance between the gripper fingers. The action space is $\mathbb{R}^4$, where the first 3 components encode the target gripper position and the last encodes the target gripper width. The block's initial position is a random location on a table. We consider predicates *NearObj* (indicates if the gripper of the arm is close to the block), *HoldingObj* (indicates if the gripper is holding the block), *LiftedObj* (indicates if the block is above the table), and *ObjAt[g]* (indicates if the block is close to a goal $g \in \mathbb{R}^3$).

We consider three specifications. First, *PickAndPlace* is

$$\phi_1 = \text{NearObj}; \text{HoldingObj}; \text{LiftedObj}; \text{ObjAt}[g],$$

where $g$ is a random goal location. Second, *PickAndPlaceStatic* is similar to the previous one, except the goal location is fixed. Third, *PickAndPlaceChoice* involves choosing between two tasks, each of which is a sequence of two subtasks similar to PickAndPlaceStatic. We learn policies using TD3 [14] with shaped rewards; each one is a fully connected NN with 2 hidden layers of 256 neurons each.

**Baselines.** We compare our approach to four state-of-the-art algorithms for learning from specifications, SPECTRL [24], QRM [19], HRM [35], and a TLTL [29] based approach, as well as a state-of-the-art hierarchical RL algorithm, R-AVI [25], that leverages state abstractions. We used publicly available implementations of SPECTRL, QRM, HRM and R-AVI. For QRM and HRM, we manually encoded the tasks as reward machines with continuous rewards. The variants QRM+CR and HRM+CR use counterfactual reasoning to reuse samples during training. Our implementation of TLTL uses the quantitative semantics defined in Li et al. [29] with ARS to learn a single policy for each task. We used the subgoal regions and the abstract graph generated by our algorithm as inputs to R-AVI. Since R-AVI only supports disjoint subgoal regions and furthermore assumes the ability to sample from any subgoal region, we only ran R-AVI on supported benchmarks. The learning curves

---

[4]This choice is the distribution of states reaching $u$ by the path policy $\pi_\rho$ eventually returned by DIRL. Thus, it ensures that the training and test distributions for edge policies in $\pi_\rho$ are equal.

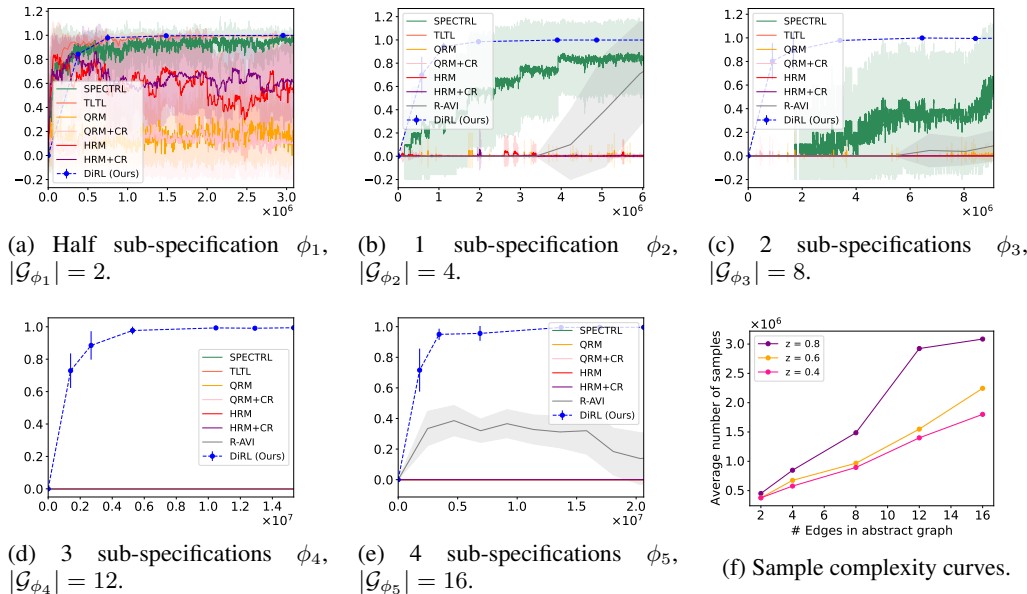

(a) Half sub-specification $\phi_1$, $|\mathcal{G}_{\phi_1}| = 2$.

(b) 1 sub-specification $\phi_2$, $|\mathcal{G}_{\phi_2}| = 4$.

(c) 2 sub-specifications $\phi_3$, $|\mathcal{G}_{\phi_3}| = 8$.

(d) 3 sub-specifications $\phi_4$, $|\mathcal{G}_{\phi_4}| = 12$.

(e) 4 sub-specifications $\phi_5$, $|\mathcal{G}_{\phi_5}| = 16$.

(f) Sample complexity curves.

Figure 3: (a)-(e) Learning curves for 16-Rooms environment with different specifications increasing in complexity from from (a) to (e). $x$-axis denotes the number of samples (steps) and $y$-axis denotes the estimated probability of success. Results are averaged over 10 runs with error bars indicating $\pm$ standard deviation. (f) shows the average number of samples (steps) needed to achieve a success probability $\geq z$ ($y$-axis) as a function of the size of the abstract graph $|\mathcal{G}_\phi|$.

for R-AVI denote the probability of reaching the final goal region in the $y$-axis which is an upper bound on the probability of satisfying the specification. Note that DIRL returns a policy only after the search finishes. Thus, to plot learning curves, we ran our algorithm multiple times with different number of episodes used for learning edge policies.

**Results.** Figure 3 shows learning curves on the specifications for 16-Rooms environment with all doors open. None of the baselines scale beyond $\phi_2$ (one segment), while DIRL quickly converges to high-quality policies for all specifications. The TLTL baseline performs poorly since most of these tasks require stateful policies, which it does not support. Though SPECTRL can learn stateful policies, it scales poorly since (i) it does not decompose the learning problem into simpler ones, and (ii) it does not integrate model-based planning at the high-level. Reward Machine based approaches (QRM and HRM) are also unable to handle complex specifications, likely because they are completely based on model-free RL, and do not employ model-based planning at the high-level. Although R-AVI uses model-based planning at the high-level in conjunction with low-level RL, it does not scale to complex specifications since it trains all edge policies multiple times (across multiple iterations) with different initial state distributions; in contrast, our approach trains any edge policy at most once.

We summarize the scalability of DIRL in Figure 3f, where we show the average number of steps needed to achieve a given success probability $z$ as a function of the number of edges in $\mathcal{G}_\phi$ (denoted by $|\mathcal{G}_\phi|$). As can be seen, the sample complexity of DIRL scales roughly linearly in the graph size. Intuitively, each subtask takes a constant number of steps to learn, so the total number of steps required is proportional to $|\mathcal{G}_\phi|$. In the supplement, we show learning curves for 9-Rooms (Figure 6) for a variety of specifications, and learning curves for a variant of 16-Rooms with many blocked doors with the same specifications described above (Figure 7). These experiments demonstrate the robustness of our tool on different specifications and environments. For instance, in the 16-Rooms environment with blocked doors, fewer policies satisfy the specification, which makes learning more challenging but DIRL is still able to learn high-quality policies for all the specifications.

Next, we show results for the Fetch environment in Figure 4. The trends are similar to before— DIRL leverages compositionality to quickly learn effective policies, whereas the baselines are ineffective. The last task is especially challenging, taking DIRL somewhat longer to solve, but it ultimately achieves similar effectiveness. These results demonstrate that DIRL can scale to complex specifications even in challenging environments with high-dimensional state spaces.

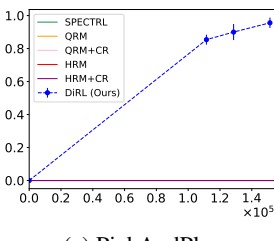
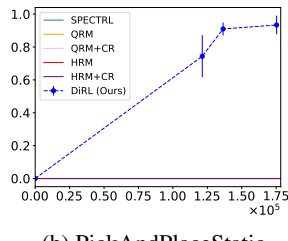
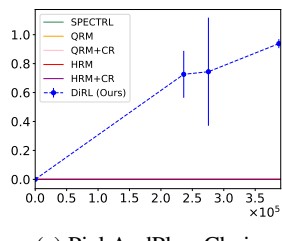

(a) PickAndPlace  (b) PickAndPlaceStatic  (c) PickAndPlaceChoice

Figure 4: Learning curves for Fetch environment; $x$-axis denotes the total number of samples (steps) and $y$-axis denotes the estimated probability of success. Results are averaged over 5 runs with error bars indicating $\pm$ standard deviation.

## 6   Conclusions

We have proposed DIRL, a reinforcement learning approach for logical specifications that leverages the compositional structure of the specification to decouple high-level planning and low-level control. Our experiments demonstrate that DIRL can effectively solve complex continuous control tasks, significantly improving over existing approaches. Logical specifications are a promising approach to enable users to more effectively specify robotics tasks; by enabling more scalable learning of these specifications, we are directly enabling users to specify more complex objectives through the underlying specification language. While we have focused on SPECTRL specifications, we believe our approach can also enable the incorporation of more sophisticated features into the underlying language, such as conditionals (i.e., only perform a subtask upon observing some property of the environment) and iterations (i.e., repeat a subtask until some objective is met).

**Limitations**

DIRL assumes the ability to sample trajectories starting at any state $s \in S$ that has been observed before, whereas in some cases it might only be possible to obtain trajectories starting at some initial state. One way to overcome this limitation is to use the learnt path policies for sampling—i.e., in order to sample a state from a subgoal region $\beta(u)$ corresponding to a vertex $u$ in the abstract graph, we could sample an initial state $s_0 \sim \eta$ from $\beta(u_0)$ and execute the path policy $\pi_{\rho_u}$ corresponding to the shortest path $\rho_u$ from $u_0$ to $u$ starting at $s_0$. Upon successfully reaching $\beta(u)$ (we can restart the sampling procedure if $\beta(u)$ is not reached), the system will be in a state $s \sim \eta_u$ in $\beta(u)$ from which we can simulate the system further.

Another limitation of our approach is that we only consider path policies. It is possible that an optimal policy must follow different high-level plans from different states within the same subgoal region. We believe this limitation can be addressed in future work by modifying our algorithm appropriately.

**Societal impacts**

Our work seeks to improve reinforcement learning for complex long-horizon tasks. Any progress in this direction would enable robotics applications both with positive impact—e.g., flexible and general-purpose manufacturing robotics, robots for achieving agricultural tasks, and robots that can be used to perform household chores—and with negative or controversial impact—e.g., military applications. These issues are inherent in all work seeking to improve the abilities of robots.

**Acknowledgements and Funding**

We thank the anonymous reviewers for their helpful comments. Funding in direct support of this work: CRA/NSF Computing Innovations Fellow Award, DARPA Assured Autonomy project under Contract No. FA8750-18-C-0090, ONR award N00014-20-1-2115, NSF grant CCF-1910769, and ARO grant W911NF-20-1-0080.

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
