# A Reduction to Abstract Reachability

In this section, we detail the construction of the abstract graph $\mathcal{G}_\phi$ from a SPECTRL specification $\phi$. Given two sets of finite trajectories $\mathcal{Z}_1, \mathcal{Z}_2 \subseteq \mathcal{Z}_f$, let us denote by $\mathcal{Z}_1 \circ \mathcal{Z}_2$ the concatenation of the two sets—i.e.,

$$\mathcal{Z}_1 \circ \mathcal{Z}_2 = \left\{ \zeta \in \mathcal{Z}_f \ \middle| \ \begin{array}{c} \exists i < t \ . \ \zeta_{0:i} \in \mathcal{Z}_1 \\ \wedge \ \zeta_{(i+1):t} \in \mathcal{Z}_2 \end{array} \right\}.$$

In addition to the abstract graph $\mathcal{G} = (U, E, u_0, F, \beta, \mathcal{Z}_{\text{safe}})$ we also construct a set of safe terminal trajectories $\mathcal{Z}_{\text{term}} = \bigcup_{u \in F} \mathcal{Z}_{\text{term}}^u$ where $\mathcal{Z}_{\text{term}}^u \subseteq \mathcal{Z}_f$ is the set of terminal trajectories for the final vertex $u \in F$. Now, we define what it means for a finite trajectory $\zeta$ to satisfy the pair $(\mathcal{G}, \mathcal{Z}_{\text{term}})$.

**Definition A.1.** A finite trajectory $\zeta = s_0 \xrightarrow{a_0} s_1 \xrightarrow{a_1} \cdots \xrightarrow{a_{t-1}} s_t$ in $\mathcal{M}$ satisfies the pair $(\mathcal{G}, \mathcal{Z}_{\text{term}})$ (denoted $\zeta \models (\mathcal{G}, \mathcal{Z}_{\text{term}})$) if there is a sequence of indices $0 = i_0 \leq i_1 < \cdots < i_k \leq t$ and a path $\rho = u_0 \to u_1 \to \cdots \to u_k$ in $\mathcal{G}$ such that

- $u_k \in F$,
- for all $j \in \{0, \ldots, k\}$, we have $s_{i_j} \in \beta(u_j)$,
- for all $j < k$, letting $e_j = u_j \to u_{j+1}$, we have $\zeta_{i_j:i_{j+1}} \in \mathcal{Z}_{\text{safe}}^{e_j}$, and
- $\zeta_{i_k:t} \in \mathcal{Z}_{\text{term}}^{u_k}$.

We now outline the inductive construction of the pair $(\mathcal{G}_\phi, \mathcal{Z}_{\text{term},\phi})$ from a specification $\phi$ such that any finite trajectory $\zeta \in \mathcal{Z}_f$ satisfies $\phi$ if and only if $\zeta$ satisfies $(\mathcal{G}_\phi, \mathcal{Z}_{\text{term},\phi})$.

**Objectives** ($\phi = \texttt{achieve } b$). The abstract graph is $\mathcal{G}_\phi = (U, E, u_0, F, \beta, \mathcal{Z}_{\text{safe}})$ where

- $U = \{u_0, u_b\}$ with $\beta(u_0) = S$ and $\beta(u_b) = S_b = \{s \mid s \models b\}$,
- $E = \{u_0 \to u_b\}$,
- $F = \{u_b\}$ and,
- $\mathcal{Z}_{\text{safe}}^{(u_0, u_b)} = \mathcal{Z}_{\text{term}}^{u_b} = \mathcal{Z}_f$.

**Constraints** ($\phi = \phi_1 \texttt{ ensuring } b$). Let the abstract graph for $\phi_1$ be $\mathcal{G}_{\phi_1} = (U_1, E_1, u_0^1, F_1, \beta_1, \mathcal{Z}_{\text{safe},1})$ and the terminal trajectories be $\mathcal{Z}_{\text{term},1}$. Then, the abstract graph for $\phi$ is $\mathcal{G}_\phi = (U, E, u_0, F, \beta, \mathcal{Z}_{\text{safe}})$ where

- $U = U_1$, $u_0 = u_0^1$, $E = E_1$ and $F = F_1$.
- $\beta(u) = \beta_1(u) \cap S_b$ for all $u \in U \setminus \{u_0\}$ where $S_b = \{s \mid s \models b\}$, and $\beta(u_0) = S$.
- $\mathcal{Z}_{\text{safe}}^e = \mathcal{Z}_{\text{safe},1}^e \cap \mathcal{Z}_b$ for all $e \in E$ where

$$\mathcal{Z}_b = \{\zeta \in \mathcal{Z}_f \mid \forall i \ . \ s_i \models b\}.$$

- $\mathcal{Z}_{\text{term}}^u = \mathcal{Z}_{\text{term},1}^u \cap \mathcal{Z}_b$ for all $u \in F$.

**Sequencing** ($\phi = \phi_1; \phi_2$). Let the abstract graph for $\phi_i$ be $\mathcal{G}_{\phi_i} = (U_i, E_i, u_0^i, F_i, \beta_i, \mathcal{Z}_{\text{safe},i})$ and the terminal trajectories be $\mathcal{Z}_{\text{term},i}$ for $i \in \{1, 2\}$. The abstract graph $\mathcal{G}_\phi = (U, E, u_0, F, \beta, \mathcal{Z}_{\text{safe}})$ is constructed as follows.

- $U = U_1 \sqcup U_2 \setminus \{u_0^2\}$.
- $E = E_1 \sqcup E_2' \sqcup E_{1\to2}$ where

$$E_2' = \{u \to u' \in E_2 \mid u \neq u_0^2\} \quad \text{and}$$

$$E_{1\to2} = \{u^1 \to u^2 \mid u^1 \in F_1 \ \& \ u_0^2 \to u^2 \in E_2\}.$$

- $u_0 = u_0^1$ and $F = F_2$.
- $\beta(u) = \beta_i(u)$ for all $u \in U_i$ and $i \in \{1, 2\}$.
- The safe trajectories are given by

- $\mathcal{Z}^e_{\text{safe}} = \mathcal{Z}^e_{\text{safe},1}$ for all $e \in E_1$,
- $\mathcal{Z}^e_{\text{safe}} = \mathcal{Z}^e_{\text{safe},2}$ for all $e \in E'_2$ and,
- $\mathcal{Z}^{u^1 \to u^2}_{\text{safe}} = \mathcal{Z}^{u^1}_{\text{term},1} \circ \mathcal{Z}^{u^2_0 \to u^2}_{\text{safe},2}$ for all $u^1 \to u^2 \in E_{1 \to 2}$.

- $\mathcal{Z}^u_{\text{term}} = \mathcal{Z}^u_{\text{term},2}$ for all $u \in F$.

**Choice** ($\phi = \phi_1$ or $\phi_2$). Let the abstract graph for $\phi_i$ be $\mathcal{G}_{\phi_i} = (U_i, E_i, u^i_0, F_i, \beta_i, \mathcal{Z}_{\text{safe},i})$ and the terminal trajectories be $\mathcal{Z}_{\text{term},i}$ for $i \in \{1, 2\}$. The abstract graph for $\phi$ is $\mathcal{G}_\phi = (U, E, u_0, F, \beta, \mathcal{Z}_{\text{safe}})$ where:

- $U = \left(U_1 \setminus \{u^1_0\}\right) \sqcup \left(U_2 \setminus \{u^2_0\}\right) \sqcup \{u_0\}$.

- $E = E'_1 \sqcup E'_2 \sqcup E_0$ where

$$E'_i = \{u \to u' \in E_i \mid u \neq u^i_0\} \quad \text{and}$$

$$E_0 = \{u_0 \to u^i \mid i \in \{1,2\} \ \& \ u^i_0 \to u^i \in E_i\}.$$

- $F = F_1 \sqcup F_2$.
- $\beta(u) = \beta_i(u)$ for all $u \in U_i$, $i \in \{1, 2\}$ and $\beta(u_0) = S$.
- The safe trajectories are given by
    - $\mathcal{Z}^e_{\text{safe}} = \mathcal{Z}^e_{\text{safe},i}$ for all $e \in E'_i$ and $i \in \{1, 2\}$,
    - $\mathcal{Z}^{u_0 \to u^i}_{\text{safe}} = \mathcal{Z}^{u^i_0 \to u^i}_{\text{safe},i}$ for all $u_0 \to u^i \in E_0$ with $u^i \in U_i$.
- $\mathcal{Z}^u_{\text{term}} = \mathcal{Z}^u_{\text{term},i}$ for all $u \in F_i$ and $i \in \{1, 2\}$.

The constructed pair $(\mathcal{G}_\phi, \mathcal{Z}_{\text{term},\phi})$ has the following important properties.

**Lemma A.2.** *For any* SPECTRL *specification $\phi$, the following hold.*

- *For any finite trajectory $\zeta \in \mathcal{Z}_f$, $\zeta \models \phi$ if and only if $\zeta \models (\mathcal{G}_\phi, \mathcal{Z}_{term,\phi})$.*

- *For any final vertex $u$ of $\mathcal{G}_\phi$ and any state $s \in \beta(u)$, the length-1 trajectory $\zeta = s$ is contained in $\mathcal{Z}^u_{term,\phi}$.*

*Proof.* Follows from the above construction by structural induction on $\phi$. $\qquad\square$

*Proof of Theorem 3.4.* Let $\zeta = s_0 \xrightarrow{a_0} s_1 \xrightarrow{a_1} \cdots$ be an infinite trajectory. First we show that $\zeta \models \phi$ if and only if $\zeta \models \mathcal{G}_\phi$.

($\implies$) Suppose $\zeta \models \phi$. Then, there is a $t \geq 0$ such that $\zeta_{0:t} \models \phi$. From Lemma A.2, we get that $\zeta_{0:t} \models (\mathcal{G}_\phi, \mathcal{Z}_{\text{term},\phi})$ which implies that $\zeta \models \mathcal{G}_\phi$.

($\impliedby$) Suppose $\zeta \models \mathcal{G}_\phi$. Then, let $0 = i_0 \leq i_1 < \cdots < i_k$ be a sequence of indices realizing a path $u_0 \to \cdots \to u_k$ to a final vertex $u_k$ in $\mathcal{G}_\phi$. Since $s_{i_k} \in \beta(u_k)$, from Lemma A.2 we have $\zeta_{i_k:i_k} \in \mathcal{Z}^{u_k}_{\text{term},\phi}$ and hence $\zeta_{0:i_k} \models (\mathcal{G}_\phi, \mathcal{Z}_{\text{term},\phi})$. From Lemma A.2 we conclude that $\zeta_{0:i_k} \models \phi$ and therefore $\zeta \models \phi$.

Next, it follows by a straightforward induction on $\phi$ that the number of vertices in $\mathcal{G}_\phi$ is at most $|\phi| + 1$ where $|\phi|$ is the number of operators (`achieve`, `ensuring`, `;`, `or`) in $\phi$. $\qquad\square$

# B  Shaped Rewards for Learning Policies

To improve learning, we use shaped rewards for learning each edge policy $\pi_e$. To enable reward shaping, we assume that the atomic predicates additionally have a *quantitative semantics*—i.e., each atomic predicate $p \in \mathcal{P}_0$ is associated with a function $[\![p]\!]_q : S \to \mathbb{R}$. To ensure compatibility with the Boolean semantics, we assume that

$$[\![p]\!](s) = \left([\![p]\!]_q(s) > 0\right). \tag{1}$$

For example, given a state $s \in S$, the atomic predicate

$$[\![\texttt{reach } s]\!]_q(s') = 1 - \|s' - s\|$$

indicates whether the system is in a state near $s$ w.r.t. some norm $\|\cdot\|$. In addition, we can extend the quantitative semantics to predicates $b \in \mathcal{P}$ by recursively defining $[\![b_1 \wedge b_2]\!]_q(s) = \min\{[\![b_1]\!]_q(s), [\![b_2]\!]_q(s)\}$ and $[\![b_1 \vee b_2]\!]_q(s) = \max\{[\![b_1]\!]_q(s), [\![b_2]\!]_q(s)\}$. These definitions are a standard extension of Boolean logic to real values. In particular, they preserve (1)—i.e., $b \models s$ if and only if $[\![b]\!]_q(s) > 0$.

In addition to quantitative semantics, we make use of the following property to define shaped rewards.

**Lemma B.1.** *The abstract graph $\mathcal{G}_\phi = (U, E, u_0, F, \beta, \mathcal{Z}_{safe})$ of a specification $\phi$ satisfies the following:*

- *For every non-initial vertex $u \in U \setminus \{u_0\}$, there is a predicate $b \in \mathcal{P}$ such that $\beta(u) = S_b = \{s \mid s \models b\}$.*

- *For every $e \in E$, either $\mathcal{Z}_{safe}^e = \mathcal{Z}_b = \{\zeta \in \mathcal{Z} \mid \forall i \,.\, s_i \models b\}$ for some $b \in \mathcal{P}$ or $\mathcal{Z}_{safe}^e = \mathcal{Z}_{b_1} \circ \mathcal{Z}_{b_2}$ for some $b_1, b_2 \in \mathcal{P}$.*

*Proof sketch.* We prove a stronger property that, in addition to the above, requires that for any $e = u_0 \to u \in E$, $\mathcal{Z}_{safe}^e = \mathcal{Z}_b$ for some $b \in \mathcal{P}$ and for any final vertex $u$, $\mathcal{Z}_{\text{term},\phi}^u = \mathcal{Z}_b$ for some $b \in \mathcal{P}$. This stronger property follows from a straightforward induction on $\phi$. $\qquad\square$

Next, we describe the shaped rewards we use to learn an edge $e = u \to u'$ in $\mathcal{G}_\phi$, which have the form

$$R_{\text{step}}(s, a, s') = R_{\text{reach}}(s, a, s') + R_{\text{safe}}(s, a, s').$$

Intuitively, the first term encodes a reward for reaching $\beta(u')$, and the second term encodes a reward for maintaining safety. By Lemma B.1, $\beta(u') = S_b$ for some $b \in \mathcal{P}$. Then, we define

$$R_{\text{reach}}(s, a, s') = [\![b]\!]_q(s').$$

The safety reward is defined by

$$R_{\text{safe}}(s, a, s') = \begin{cases} \min\{0, [\![b]\!]_q(s')\} & \text{if } \mathcal{Z}_{safe}^e = \mathcal{Z}_b \\ \min\{0, [\![b \vee b']\!]_q(s')\} & \text{if } \mathcal{Z}_{safe}^e = \mathcal{Z}_b \circ \mathcal{Z}_{b'} \wedge \psi_b \\ \min\{0, [\![b']\!]_q(s')\} & \text{if } \mathcal{Z}_{safe}^e = \mathcal{Z}_b \circ \mathcal{Z}_{b'} \wedge \neg\psi_b. \end{cases}$$

Here, $\psi_b$ is internal state keeping track of whether $b$ has held so far—i.e., $\psi_b \leftarrow \psi_b \wedge [\![b]\!](s)$ at state $s$. Intuitively, the first case is the simpler case, which checks if every state in the trajectory satisfies $b$, and the latter two cases handle a sequence where $b$ should hold for the first part of the trajectory, and $b'$ should hold for the remainder.

## C  Proof of Theorem 4.2

*Proof.* Let the abstract graph be $\mathcal{G} = (U, E, u_0, F, \beta, \mathcal{Z}_{\text{safe}})$. Let us first define what it means for a rollout to achieve a path in $\mathcal{G}$.

**Definition C.1.** *We say that an infinite trajectory $\zeta$ achieves the path $\rho$ (denoted $\zeta \models \rho$) if $\zeta \models \mathcal{G}_\rho$ where $\mathcal{G}_\rho = (U_\rho, E_\rho, u_0, \{u_k\}, \beta \downarrow \rho, \mathcal{Z}_{safe} \downarrow_\rho)$ with $U_\rho = \{u_j \mid 0 \leq j \leq k\}$, $E_\rho = \{u_j \to u_{j+1} \mid 0 \leq j < k\}$ and $\beta \downarrow \rho$ and $\mathcal{Z}_{safe} \downarrow_\rho$ are $\beta$ and $\mathcal{Z}_{safe}$ restricted to the vertices and the edges of $\mathcal{G}_\rho$, respectively.*

From the definition it is clear that for any infinite trajectory $\zeta$, if $\zeta \models \rho$ then $\zeta \models \mathcal{G}$ and therefore

$$\Pr_{\zeta \sim \mathcal{D}_{\pi_\rho}} [\zeta \models \mathcal{G}] \geq \Pr_{\zeta \sim \mathcal{D}_{\pi_\rho}} [\zeta \models \rho]. \tag{2}$$

Let us now define a slightly stronger notion of achieving an edge.

**Definition C.2.** *An infinite trajectory $\zeta = s_0 \to s_1 \to \cdots$ is said to greedily achieve the path $\rho$ (denoted $\zeta \models_g \rho$) if there is a sequence of indices $0 = i_0 \leq i_1 < \cdots < i_k$ such that for all $j < k$,*

- $\zeta_{i_j:\infty} \models e_j = u_j \to u_{j+1}$ *and,*

- $i_{j+1} = i(\zeta_{i_j:\infty}, e_j)$,

*where $\zeta_{i_j:\infty} = s_{i_j} \to s_{i_j+1} \to \cdots$.*

That is, $\zeta \models_g \rho$ if a partition of $\zeta$ realizing $\rho$ can be be constructed greedily by picking $i_{j+1}$ to be the smallest index $i \geq i_j$ (strictly bigger if $j > 0$) such that $s_i \in \beta(u_{j+1})$ and $\zeta_{i_j:i} \in \mathcal{Z}_{\text{safe}}^{e_j}$. Since $\zeta \models_g \rho$ implies $\zeta \models \rho$, we have

$$\Pr_{\zeta \sim \mathcal{D}_{\pi_\rho}} [\zeta \models \rho] \geq \Pr_{\zeta \sim \mathcal{D}_{\pi_\rho}} [\zeta \models_g \rho]. \tag{3}$$

Let $\rho_{j:k}$ denote the $j$-th suffix of $\rho$. We can decompose the probability $\Pr_{\zeta \sim \mathcal{D}_{\pi_\rho}}[\zeta \models_g \rho]$ as follows.

$$
\begin{aligned}
\Pr_{\zeta \sim \mathcal{D}_{\pi_\rho}} [\zeta \models_g \rho] &= \Pr_{\zeta \sim \mathcal{D}_{\pi_\rho}} [\zeta \models e_0 \ \wedge \ \zeta_{i(\zeta, e_0):\infty} \models_g \rho_{1:k}] \\
&= \Pr_{\zeta \sim \mathcal{D}_{\pi_{e_0}}} [\zeta \models e_0] \cdot \Pr_{\zeta \sim \mathcal{D}_{\pi_\rho}} [\zeta_{i(\zeta, e_0):\infty} \models_g \rho_{1:k} \mid \zeta \models e_0] \\
&= P(e_0; \pi_{e_0}, \eta_0) \cdot \Pr_{s_0 \sim \eta_{\rho_{0:1}}, \zeta \sim \mathcal{D}_{\pi_{\rho_{1:k}}, s_0}} [\zeta \models_g \rho_{1:k}]
\end{aligned}
$$

where the last equality followed from the definition of $\eta_{\rho_{0:1}}$ and the Markov property of $\mathcal{M}$. Applying the above decomposition recursively, we get

$$
\begin{aligned}
\Pr_{\zeta \sim \mathcal{D}_{\pi_\rho}} [\zeta \models_g \rho] &= \prod_{j=0}^{k-1} P(e_j; \pi_{e_j}, \eta_{\rho_{0:j}}) \\
&= \exp(\log(\prod_{j=0}^{k-1} P(e_j; \pi_{e_j}, \eta_{\rho_{0:j}}))) \\
&= \exp(-(-\sum_{j=0}^{k-1} \log P(e_j; \pi_{e_j}, \eta_{\rho_{0:j}}))) \\
&= \exp(-c(\rho)).
\end{aligned}
$$

Therefore, from Equations 2 and 3, we get the required bound. $\qquad \square$

# D Experimental Methodology

Our tool learns the low-level NN policies for edges using an off-the-shelf RL algorithm. For the Rooms and Fetch environments, we learn policies using ARS [32] and TD3 [14] with shaped rewards, respectively.

For each specification on an environment, we first construct its abstract graph. In DIRL, each edge policy $\pi_e$ is trained using $k$ episodes of interactions with the environment. For the purpose of generating a learning curve, we run DIRL for each specification with several values of $k$. For each $k$ value, we plot the sum total of the samples taken to train all edge policies against the probability with which the computed policy reaches a final subgoal region.

For a fair comparison with the baselines, if each episode for learning an edge policy in DIRL is run for $m$ steps, we run the episodes of the baselines for $m \cdot d + c$ steps, where $d$ is the maximum path length to reach a final vertex in the abstract graph of the specification and $c > 0$ is a buffer. Intuitively, this approach ensures that all tools get a similar number of steps in each episode to learn the specification.

 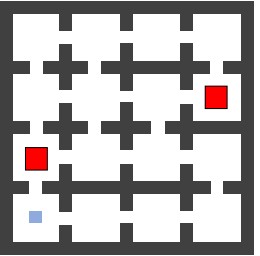

(a) 16-Rooms (All doors open)   (b) 16-Rooms (Some doors open)

Figure 5: 16-Rooms Environments. Blue square indicates the initial room. Red squares represent obstacles. (a) illustrates the segments in the specifications.

# E   Case Study: Rooms Environment

We consider environments with several interconnected rooms. The rooms are separated by thick walls and are connected through bi-directional doors.

The environments are a 9-Rooms environment, (Figure 1), a 16-Rooms environment with all doors open (Figure 5a), and 16-Rooms environment with some doors open (Figure 5b). The red blocks indicate obstacles. A robot can pass through those rooms by moving around the red blocks. The robot is initially placed randomly in the center of the room with the blue box (bottom-left corner).

Rooms are identified by the tuple $(r, c)$ denoting the room in the $r$-th row and $c$-th column. We use the convention that the bottom-left corner is room (0,0). Predicate reach $(r, c)$ is interpreted as reaching the center of the $(r, c)$-th room and predicate avoid $(r, c)$ is interpreted as avoiding the center of the $(r, c)$-th room. For clarity, we omit the word achieve from specifications of the form achieve $b$ denoting such a specification using just the predicate $b$.

## E.1   9-Rooms Environment

**Specifications.**

1. $\phi_1 := $ reach $(2, 0)$; reach $(0, 0)$

   Go to the top-left corner and then return to the bottom-left corner (initial room); red blocks not considered obstacles.

   This specification is difficult for standard RL algorithms that do not store whether the first sub-task has been achieved. In these cases, a stateless policy will not be able to determine whether to move upwards or downwards. In contrast, DIRL (as well as SPECTRL and RM based approaches) augment the state space to automatically keep track of which sub-tasks have been achieved so far.

2. $\phi_2 := $ reach $(2, 0)$ or reach $(0, 2)$

   Either go to the top-left corner or to the bottom-right corner (obstacles are not considered).

3. $\phi_3 := \phi_2$; reach $(2, 2)$

   After completing $\phi_2$, go to the top-right corner (obstacles not considered).

   This specification combines two choices of similar difficulty yet only one is favorable to fulfilling the specification since the direct path to the top-right corner from the bottom-right one is obstructed by walls.

4. $\phi_4 := $ reach $(2, 0)$ ensuring avoid $(1, 0)$

   Reach the top-left (while considering the obstacles).

5. $\phi_5 := \phi_4$ or reach $(0, 2)$; reach $(2, 2)$

   Either go to the top-left corner or bottom-right corner enroute to the top-right corner (while considering the obstacles).

   This specification is similar to $\phi_3$ except that the choices are of unequal difficulty due to the placement of the red obstacle. In this case, the non-greedy choice is favorable for completing the task.

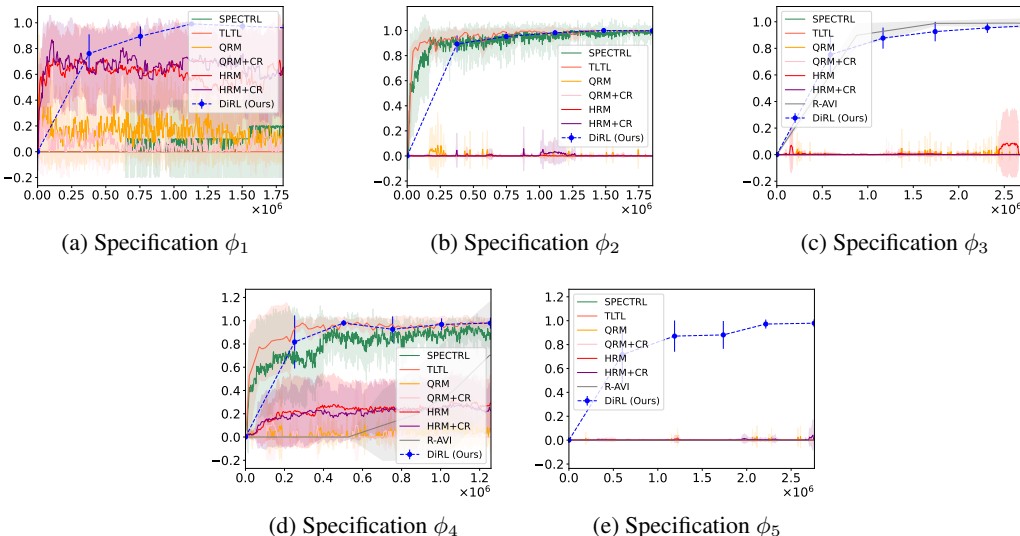

(a) Specification $\phi_1$      (b) Specification $\phi_2$      (c) Specification $\phi_3$

(d) Specification $\phi_4$      (e) Specification $\phi_5$

Figure 6: Learning curves for 9-Rooms environment with different specifications. $x$-axis denotes the number of samples (steps) and $y$-axis denotes the estimated probability of success. Results are averaged over 10 runs with error bars indicating $\pm$ standard deviation.

**Hyperparameters.** The edge policies are learned using ARS [32] (version V2-t) with neural network policies and the following hyperparameters.

- Step-size $\alpha = 0.3$.
- Standard deviation of exploration noise $\nu = 0.05$.
- Number of directions sampled per iteration is 30.
- Number of top performing directions to use $b = 15$.

To plot the learning curve, we use values of

$$k \in \{3000, 6000, 12000, 18000, 24000, 30000\}$$

where each episode consists of $m = 20$ steps.

**Results.** The learning curves for these specifications are shown in Figure 6. While most tools perform reasonably well on specifications $\phi_2$ (Figure 6b) and $\phi_4$ (Figure 6d), the baselines are unable to learn to satisfy $\phi_3$ (Figure 6c) and $\phi_5$ (Figure 6e) except for R-AVI which learns to satisfy $\phi_3$ as well.

### E.2 16-Rooms Environment

**Specifications.** We describe the five specifications used for the 16-rooms environment, which are designed to increase in difficulty. First, we define a *segment* as the following specification: Given the current location of the agent, the goal is to reach a room diagonally opposite to it by visiting at least one of the rooms at the remaining two corners of the rectangle formed by the current room and the goal room—e.g., in the 9-Rooms environment, to visit $S_3$ from the initial room, the agent must visit either $S_1$ or $S_2$ first.

Then, we design specifications of varying sizes by sequencing several segments one after the other. In the first segment, the agent's current location is the initial room. In subsequent segments, the current location is the goal room of the previous segment. In addition, the agent must always avoid the obstacles in the environment. We create five such specifications, one half-segment and specifications up to four segments ($\phi_1$ to $\phi_5$), as illustrated in Figure 5a and described below:

1. $\phi_1$ corresponds to the *half-segment* enroute (2,2) from (0,0). Thus $\phi_1$ is a choice between (0,2) and (2,0).

2. $\phi_2$ is the first segment that goes from (0,0) to (2,2)

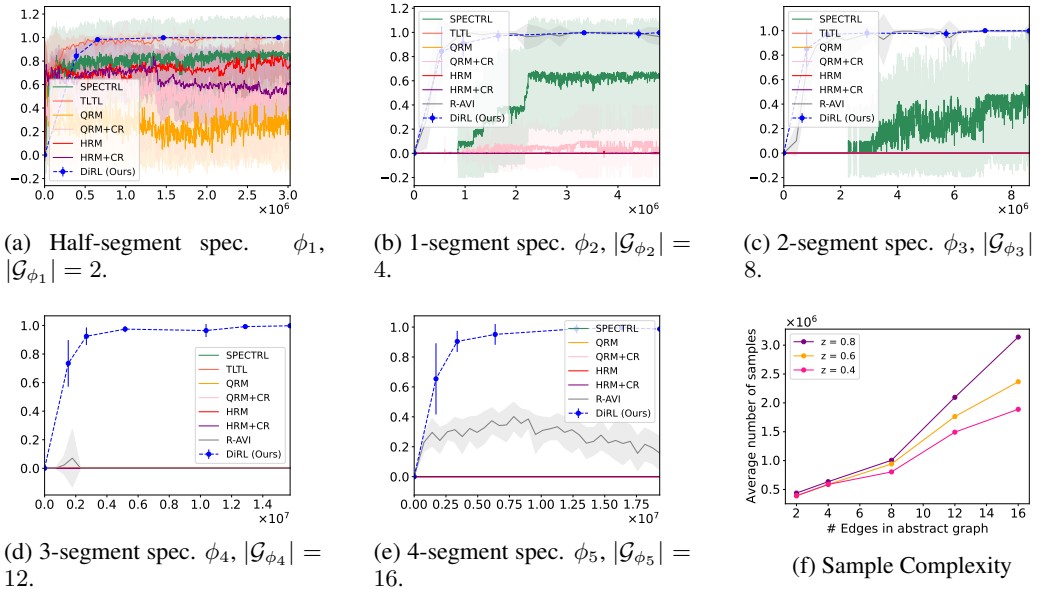

(a) Half-segment spec. $\phi_1$, $|\mathcal{G}_{\phi_1}| = 2$.

(b) 1-segment spec. $\phi_2$, $|\mathcal{G}_{\phi_2}| = 4$.

(c) 2-segment spec. $\phi_3$, $|\mathcal{G}_{\phi_3}| = 8$.

(d) 3-segment spec. $\phi_4$, $|\mathcal{G}_{\phi_4}| = 12$.

(e) 4-segment spec. $\phi_5$, $|\mathcal{G}_{\phi_5}| = 16$.

(f) Sample Complexity

Figure 7: (a)-(e) Learning curves for 16-Rooms environment with some blocked doors (Figure 5b) with different specifications increasing in complexity from (a) to (e). $x$-axis denotes the number of samples (steps) and $y$-axis denotes the estimated probability of success. Results are averaged over 10 runs with error bars indicating $\pm$ standard deviation. (f) shows the average number of samples (steps) needed to achieve a success probability $\geq z$ ($y$-axis) as a function of the size of the abstract graph $|\mathcal{G}_\phi|$.

    3. $\phi_3$ augments $\phi_2$ with a second segment to (3,1).

    4. $\phi_4$ augments $\phi_3$ with a segment to (1,3)

    5. $\phi_5$ augments $\phi_4$ with a segment to (0,1)

We denote by $|\mathcal{G}_\phi|$ the number of edges in the abstract graph corresponding to the specification $\phi$.

**Hyperparameters.** We use the same hyperparameters of ARS as the ones used for the 9-Rooms environment. We run experiments for

$$k \in \{6000, 12000, 24000, 48000, 60000, 72000\}.$$

**Results.** The learning curves for the environment with all open doors and the constrained environment with some open doors are shown in Figure 3 and Figure 7, respectively.

# F   Case Study: Fetch Environment

The fetch robotic arm from OpenAI Gym is visualized in Figure 8. Let us denote by $s_r = (s_r^x, s_r^y, s_r^z) \in \mathbb{R}^3$ the position of the gripper, $s_o \in \mathbb{R}^3$ the relative position of the object (black block) w.r.t. the gripper, $s_g \in \mathbb{R}^3$ the goal location (red sphere) and $s_w \in \mathbb{R}$ the width of the gripper. Let $c$ denote the width of the object and $z_\epsilon = (0, 0, \epsilon + c)$ for $\epsilon > 0$. Then, we define the following predicates.

- *NearObj* holds true in states in which the gripper is wide open, aligned with the object and is slightly above the object:

$$\text{NearObj}(s) = \left( \|s_o + z_\epsilon\|_2^2 + (s_w - 2c)^2 < \delta_1 \right)$$

- *HoldingObj* holds true in states in which the gripper is close to the object and its width is close to the object's width:

$$\text{HoldingObj}(s) = \left( \|s_o\|_2^2 + (s_w - c)^2 < \delta_2 \right)$$

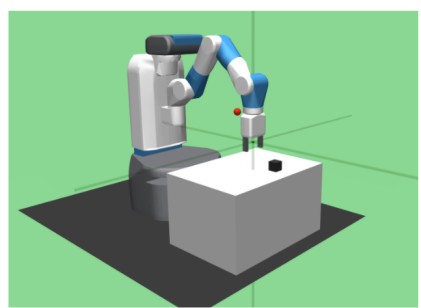

Figure 8: Fetch robotic arm.

- *LiftedObj* holds true in states in which the object is above the surface level of the table:

$$\text{LiftedObj}(s) = \left(s_r^z + s_o^z > \delta_3\right)$$

- *ObjAt*[g] holds true in states in which the object is close to $g$:

$$\text{ObjAt}[g](s) = \left(\|s_r + s_o - g\|_2^2 < \delta_4\right)$$

Then the specifications we use are the following.[5]

- PickAndPlace: $\phi_1 = $ NearObj; HoldingObj; LiftedObj; ObjAt[$s_g$].
- PickAndPlaceStatic: NearObj; HoldingObj; LiftedObj; ObjAt[$g_1$] where $g_1$ is a fixed goal.
- PickAndPlaceChoice: $\big($NearObj; HoldingObj; LiftedObj$\big)$; $\big($(ObjAt[$g_1$]; ObjAt[$g_2$]) or (ObjAt[$g_3$]; ObjAt[$g_4$])$\big)$.

**Hyperparameters.** We use TD3 [14] for learning edge policies with the following hyperparameters.

- Discount $\gamma = 0.95$.
- Adam optimizer; actor learning rate $0.0001$; critic learning rate $0.001$.
- Soft update targets $\tau = 0.005$.
- Replay buffer of size $200000$.
- $100$ training steps performed every $100$ environment steps.
- A minibatch of $256$ steps used per training step.
- Exploration using gaussian noise with $\sigma = 0.15$.

We run experiments for $k \in \{1000, 2000, 4000\}$ and each episode consists of $m = 40$ steps.

---

[5]We denote `achieve` $b$ using just the predicate $b$.