# OpenReview forum: "Compositional Reinforcement Learning from Logical Specifications"
_NeurIPS.cc/2021/Conference — NeurIPS 2021 Poster_

### Official Review · Reviewer_4uY8 · 2021-07-16

**Rating:** 6
**Confidence:** 3

**Summary:**

This paper addresses problems where the task to be accomplished is given by a specification language (Spectrl). Tasks given in this specification language can be represented as reachability problems on an abstracted graph, where nodes in the abstracted graph represent regions of the underlying state space of the original problem. To find a high-level plan to satisfy the task specification, Dijkstra's algorithm is used. To learn low-level controllers to complete each of the subtasks, reinforcement learning is used. The probability of a subtask being completed by a low-level controller is evaluated empirically, and used to update costs in the high level graph so that the lowest cost path in the high-level graph corresponds to the path maximising the probability of completing the task.

I think the paper is well written, and addresses an interesting problem. To my understanding, the general idea of combining high-level model-based planning with RL for low-level control has been addressed in existing works. However, this is the first work to do so for more complex task specifications.


**Limitations And Societal Impact:**

Some technical contribution limitations identified.

**Main Review:**

I think the authors need to be clearer about the contributions made by this paper (I would suggest that the authors explicitly enumerate their contributions in the introduction). In the introduction, the authors seem to indicate that a novelty of their work is integrating high-level model-based planning with learning of low-level control. However, this general idea has been presented in other papers such as Illanes (2020), where the high-level specification is given in PDDL (and also considers MaxProb formulations), or arguably even Eysenback (2019).

Therefore, my understanding is that while other works have considered the idea of high-level planning over subtasks + low-level controllers via reinforcement learning, this is the first work consider more powerful task specifications in the form of tasks specified by Spectrl.

Question 1).
Can the authors please compare their work with Illanes (2020). Is the main difference that Illanes (2020) considers high-level goals specified in PDDL, whereas here goals may be specified using Spectrl, or are there other significant advances made by this work?

The high-level planner operates on a deterministic graph, and outputs the maximum probability path (rather than a policy for all high-level states). Therefore, if the low-level controller fails to reach the desired sub-goal, and accidentally reaches a different sub-goal, the algorithm will not switch appropriately to a different sub-policy. Given that the empirical transition probabilities between high-level abstract states are computed empirically anyway (to compute the edge costs in the graph), I wonder why we wouldn't just treat the high-level abstract graph as an MDP, and compute a high-level policy for that MDP rather than a path? I understand that this would increase the computational burden as we have to learn sub-policies for more states, but this seems like a more natural formulation for stochastic environments, especially considering that the transition probabilities need to be empirically estimated anyway.

Question 2).
Can the authors comment on the advantages of treating the high-level planning problem as deterministic, rather than an MDP? It seems that an MDP would more naturally handle cases where the sub-policy fails to reach the desired sub-goal, and transition probabilities between sub-goals are already being computed empirically anyway.

Minor comment: It might aid in the writing of the paper by referring to the sub-policies as "options" as is often done within RL literature.

References:

Illanes, León, et al. "Symbolic plans as high-level instructions for reinforcement learning." Proceedings of the International Conference on Automated Planning and Scheduling. Vol. 30. 2020.

Eysenbach, Benjamin, Ruslan Salakhutdinov, and Sergey Levine. "Search on the replay buffer: Bridging planning and reinforcement learning." NeurIPS 2019


**Time Spent Reviewing:**

4

---

> ### Author Response · Authors · 2021-08-10
> **Response to Reviewer 4uY8**
>
> We thank the reviewer for their valuable comments. We address the reviewer's comments below:
>
> **I. Contributions:** In summary, our contributions are stated below. We will add this to the revision.
>
>    - A novel compositional algorithm to learn policies for continuous (infinite-state) domains from complex high-level specifications that interleaves high-level model-based planning with low-level RL. (clarified below)
>    -Theoretical analysis of our algorithm demonstrates that it aims to maximize a lower bound on the satisfaction probability of the specification.
>    -Empirical evaluation that demonstrates that our algorithm outperforms several state-of-the-art algorithms for learning from high-level specifications.
>
> With regards to integration of high-level model-based planning and low-level RL, prior work either first performs high-level model-based planning and then performs low-level RL [1] or first performs low-level RL and then performs high-level planning [2]. Our work interleaves both steps -- high-level planning leverages low-level RL and vice-versa -- resulting in an efficient practical algorithm.
>
> **II. Comparison with [1] (Ilanes et. al.):** We outline the important differences below:
>
>    1. **Interleaving vs. Planning guided learning:** [1] performs high-level planning to generate a plan first, and then performs low-level RL to learn policies for the options in the computed plan. The drawback of this is that if an option cannot be realized in practice, then the algorithm will not be able to compute a policy for the complete task. For example, in our motivating example from Figure 1, if the high-level plan chooses option (S1 to S2) and then option (S2 to S3) then the algorithm will fail to compute a satisfying policy since option (S2 to S3) is not realizable in practice.
>
>        Our algorithm (DiRL) overcomes this drawback by interleaving planning and learning. It uses the satisfaction probabilities of edge policies (learned using RL) in planning.
>
>    1. **Expressiveness of the input:** As pointed out by the reviewer, [1] considers reachability tasks only whereas we consider complex SpectRL specifications that can also encode safety properties.
>
>    1. **Assumptions:** The theoretical guarantees in [1] rely on the assumption that there is a policy for every option in the pre-computed high-level plan such that the individual policies reach the corresponding goal states with probability 1.  As explained in Point (1), this assumption is often violated in practice. We do not require such an assumption, instead our algorithm estimates and uses the probability of satisfaction of the options (edge policies) in planning.
>
> **III. Comparison with [2] (Eysenbach et. al.):** We outline the important differences below.
>
>    1. **Expressiveness:** [2] considers pure reachability tasks whereas we consider complex tasks specified in SpectRL. For example, they do not handle safety constraints in their algorithm.
>
>    1. **Algorithm:** [2] uses goal-conditioned RL to first learn a goal-conditioned policy and then uses planning to compute a path from the current state to the goal state. It does not use planning to guide low-level RL whereas DiRL uses planning to determine what edge to learn next, thereby learning to solve the task without necessarily learning all edge policies.
>
> **IV. High-level Model: Deterministic vs. MDP:** The benefits of the deterministic model are that it is much simpler to plan on and also gives us a natural interleaving of Dijkstra with low-level learning that seems to work well in practice. Furthermore, our model also enables us to learn path policies for a specification without necessarily having to learn all edge policies. Also, we encode the probability of achieving an edge into the costs of the high level model and therefore, some probabilistic information is incorporated into planning.
>
> We agree with the reviewer that considering probabilistic models for the high level model is an interesting direction for future work and as suggested this would help in cases where the robot needs to recover from failure of an edge in the best path. As mentioned in our limitations, the restriction to path policies - where failure recovery cannot occur - is the main disadvantage of our model and we hope to improve on this in future work. Using probabilistic high-level models presents interesting technical challenges. For instance, it will be tricky to deal with overlapping subgoal regions as the reach probabilities do not sum up to one.
>
> Finally, the work [3] suggested by Reviewer YbXN does in fact use a probabilistic model at the high level and uses value iteration to plan in this model. Our initial experiments demonstrate that our algorithm requires fewer samples in comparison to [3]. For the motivating example in Figure 1, DiRL requires ~$2\times10^6$ samples as opposed to ~$5\times10^6$ samples required by [3]. We expect to see similar trends on other benchmarks as well. We will add a comparison in the revision.
>
> **References:**
> [1] Illanes, León, et al. "Symbolic plans as high-level instructions for reinforcement learning." Proceedings of the International Conference on Automated Planning and Scheduling. Vol. 30. 2020.
> [2] Eysenbach, Benjamin, Ruslan Salakhutdinov, and Sergey Levine. "Search on the replay buffer: Bridging planning and reinforcement learning."
> [3]: "Abstract Value Iteration for Hierarchical Reinforcement Learning", Kishor Jothimurugan,  Osbert Bastani, and Rajeev Alur.

---

> > ### Comment · Reviewer_4uY8 · 2021-08-26
> > **Response response**
> >
> > Thanks for your detailed and clear response. It has certainly aided in my understanding of your paper, and leads me to view it increasingly positively.

---

### Official Review · Reviewer_yVhC · 2021-07-16

**Rating:** 6
**Confidence:** 3

**Summary:**

The paper proposes a framework for combining RL with logical specifications. In particular, the task is specified using local statements, which are then converted into a graph and high-level planning (using Dijkstra's algorithm) is used to attain the goal while adhering to any safety constraints. Within this graph, nodes are sets of low-level states, while edges represent the probability of a policy (learned using model-free RL) transitioning between nodes. While there exist other approaches that combine logical (e.g. LTL) specifications with RL, experiments in two continuous control domains demonstrate that the method outperforms these existing approaches, and scales only linearly with the complexity of the specification.


**Limitations And Societal Impact:**

While the paper discusses a few of its limitations, if I'm understanding correctly, then one additional limitation is that a mapping from states to predicates be given upfront, and all predicates (nodes) be known before learning begins. This requires human expertise, and may be very hard for complex tasks.


**Main Review:**

The paper tackles the problem of solving tasks whose goals and constraints are specified using logical predicates, instead of reward functions. I feel this is an important direction (since the ability to encode goals and safety constraints in a human-understandable form will be critical to real-world applications) and, to the best of my knowledge, the paper makes a novel and interesting contribution towards this. Although the paper was generally well written, I found that certain parts were very hard to follow due to space constraints (reading the supplementary material helped, but it would be better if the main text were more self-contained). I have a few questions to help with clarification, and also one concern about how the graph is constructed in the first place, which I'll outline below.

Main comments

1. One aspect I'm not clear on is the grounding of predicates to MDP states. How exactly is the link between predicates and low-level states established?. Line 177 implies that this mapping needs to be done manually. If so, is this then akin to a human designer manually specifying a symbolic representation (which could be hard for very complex domains) and then learning how the symbolic states are connected? If so, this seems like a fair amount of information that must be encoded manually (other approaches like [1] can autonomously construct them). Are the other baselines provided with similar information?
2. A follow-up question is the following: since the subgoal regions can contain overlapping states, we could have a low-level state s be in both nodes A and B. If we are learning a low-level policy to reach A, but the agent observes state s, how could it be certain that it has not reached B instead?
3. I found that I couldn't follow Section 5 without the supplementary material at all. Terms like "sub-specification" and "half sub-specification" were completely opaque to me, but the longer explanation in the appendix made sense. If possible, more detail could be incorporated into the main text (I feel like a few of the definitions in Section 4.2 could be removed to make room).
4. I don't quite follow why on Line 114 the policy is not the regular Markov policy, but rather a function of the history. My understanding is that each edge policy is Markov and acts according to the MDP state, while the path policy could act based on the current node only. Any clarification would be appreciated.

Minor comments

1. In the absence of rewards, it makes sense that the agent should seek a maximally likely policy that reaches the goal. However, are there use cases where this wouldn't work? For example, in a tabular deterministic shortest path setting, all policies are equally capable of reaching the goal with probability 1 (but may take varying amounts of time). Is there any way this can be overcome?
2. The blue/grey colours in Figure 1 (left) are a bit faint and hard to pick up on, especially when printed.
3. Line 153 cyclic -> acyclic
4. Line 164: What is s_{i_j}?

[1] Pacheck, Adam, George Konidaris, and Hadas Kress-Gazit. "Automatic Encoding and Repair of Reactive High-Level Tasks with Learned Abstract Representations." Robotics Research: the 18th Annual Symposium. 2019.


****** POST REBUTTAL ******

Thank you for the response, which has helped clear up the questions and misconceptions that I had. As one of the other reviewers recommended, emphasising the exact novel contributions of the work in the introduction would be extremely helpful for an updated version.




**Time Spent Reviewing:**

6

---

> ### Author Response · Authors · 2021-08-10
> **Response to Reviewer yVhC**
>
> We thank the reviewer for the comments. We will incorporate their suggestions into our revision. We address their main comments below:
>
> **I. Grounding of Predicates in Baselines:** Yes, all baselines require that the grounding of predicates/propositions to MDP states is manually provided in order to interpret the specification. This grounding/mapping is provided symbolically. More specifically, every predicate/proposition is a function that maps MDP states to either True or False depending on whether the predicate holds at that MDP state.  This is a standard assumption in all existing work on RL from specifications.
>
> In practice, several alternatives are possible. For instance, rather than provide the predicate semantics, the user can instead provide examples of when the predicate is/isn’t satisfied, in which case we can learn a classifier to serve as the predicate. Such an approach is beyond the scope of our work, but is an interesting direction for future work, and we will add a discussion. We thank the reviewer for pointing us to [1] that automates the grounding of predicates/propositions.  Such approaches could be a valuable addition to the literature on RL from specifications.
>
> Our work also involves another mapping: Between predicates and subgoal regions. As explained in our response to Reviewer YbXN, this mapping is NOT provided manually but is inferred by the algorithm.
>
> **II. Overlapping MDP states:** As pointed out by the reviewer, we do not require the assumption that the subgoal regions be disjoint. Specifically, it is possible that a low-level MDP state is present in multiple subgoal regions.
>
> Nevertheless, having overlapping subgoal regions is not a hindrance to training a low-level edge policy. This is because each edge policy is trained to achieve a single edge with a single subgoal region (and some safety constraints). As long as the edge policy learns to reach the target subgoal region, the policy learns to achieve the edge. In the process, it is possible that the edge policy may encounter MDP states that are present in another subgoal region. However, since every edge is associated with a unique subgoal region, visiting other subgoal regions in the process is not acknowledged/detected. Specifically, for overlapping states, their presence in other subgoal regions is not acknowledged.
>
> Similarly, having overlapping subgoal regions is not a hindrance to executing a path policy. At any point, the path policy executes a unique edge policy. The path policy switches to the next edge policy only after reaching the subgoal region corresponding to the current edge. The current edge policy may reach other subgoal regions along the way but the path policy doesn’t detect/acknowledge that.
>
> **III. Non-Markovian Policies:** Each edge policy is Markovian. However, the overall path policy may be non-Markovian as it keeps track of which edge is being executed. For example, the robot could be in the same MDP state while performing two different edge tasks along the same path. In such cases, it needs to keep track of which edge it is currently trying to achieve in order to take the appropriate action in the current MDP state.
>
> As a concrete example, consider Specification 1 ($\psi_1$) in Section E.1 of the supplemental material. In this case, the agent moves upwards or downwards from an MDP state depending on which edge of the abstract graph it is trying to achieve.
>
> **IV. Minor Comments:**
>
> **1.** While our current algorithm does optimize quantitative objectives such as time, we believe it can be extended to incorporate such objectives. One approach would be to modify the abstract reachability problem into a multi-objective optimization problem which would require optimizing both the probability of satisfying the specification and the quantitative objective. This is certainly a great direction for further expansion of the expressiveness of high-level specifications.
>
> **4.** $s_{i_j}$ is the ${i_j}$-th state in the trajectory $\zeta = s_1\to s_2\to\cdots \to s_t$ where $i_j$ marks the index where $j$-th task begins.
>
> **References:**
> [1] Pacheck, Adam, George Konidaris, and Hadas Kress-Gazit. "Automatic Encoding and Repair of Reactive High-Level Tasks with Learned Abstract Representations." Robotics Research: the 18th Annual Symposium. 2019.

---

### Official Review · Reviewer_WvH4 · 2021-07-16

**Rating:** 7
**Confidence:** 4

**Summary:**

This paper addresses the problem of synthesizing (using reinforcement
learning) a policy that satisfies a logical specification. The key
insight in this, and related works, is that the implicit structure
of the specification can be used to decompose the learning problem
into several, ideally simpler, learning problems.

Technically, this work operates by first syntactically restricting the
space of (infinite horizon) tasks. This class is fairly expressive and
has the important property that tasks can be abstractly monitored
using a *finite* DAG - with (abstract) satisfaction now being a path
finding problem on the DAG abstraction. The conversion is automatic
and the complexity grows linearly in the syntactic structure.

The graph structure then admits a nature mechanism to split the task
into sub-tasks, corresponding to traversing an edge in the DAG. Thus,
by leveraging classic shortest path finding algorithms, where the cost
is determined by the traversal probability - which is itself
determined by training a policy in the workspace and empirical
evaluation - one obtains a hierarchical policy. This defines an iterative algorithm which the paper shows
can be used to increase sample efficiency compared to various baselines.



**Limitations And Societal Impact:**

I believe that this line of research, in particular incorporating high
level reasoning based on logical properties, is an important
ingredient safe AI.

Regarding the limitations, I was pleased to see that the major
limitation I had noting when reading the paper was explicitly
addressed in the conclusion. Namely, in order to trade edge policies
it is important to be able to reliably initialize the system to an
arbitrary state, or a similar assumption. While in many domains, e.g.,
simulation, this may be reasonable, in other complex domains this
presents a non-trivial problem and sub-task of its own. Nevertheless,
I feel that this work is still valuable and the above limitation could
be overcome.


**Main Review:**

At a high level, I found the writing and the exposition of the topics
well done. Furthermore, I feel that the topic of using logical
formalisms for defining tasks in RL is a rich and important topic.
This work illustrates, for example, tasks can be naturally decomposed
increasing the efficiency of learning - and perhaps even the
interpretability and verifiability of the agent.

## Empirical Evaluation
I think the authors have done a fairly good empirical evaluation. I
particularly appreciated the inclusion of the two reward machine based
algorithms, which I would have a-priori considered to be natural
comparisons. One missing comparison I would have liked to see, but am
ok not having is an off-the-shelf hierarchical RL algorithm where the
specification DAG is treated as an MDP. This would have given me a
better sense if the primary contribution is the decomposition of the
problem or the shortest path training scheme. That said, I am fairly
confident this approach would outperform such a base-line.

## Clarification about Path policies
My understanding is that it may not always be the case that an edge is
successfully traversed in the DAG abstraction. One thing I would like
clarified is what the high level polices do in such cases. The path
policies defined on line 238 seem to suggest that a path is first
selected and then the system updates the path index. But if the edge
traversal fails, is this a stuttering transition? Or is it just that
it takes an unbounded (possibly infinite) amount of time?  For example,
if the agent accidentally enters a sink in the underlying MDP? While I think
I understand how to fill in these holes, a more explicit discussion would
have been appreciated.

## Role of reward shaping
The class of tasks considered in inherently sparse. The authors
propose a reward shaping scheme, but I can't help but wonder how such
an approach would compare with recent advances in learning directly
from sparse rewards, e.g., Hindsight Experience Replay. It seems that
one of the reasons splitting up the task would improve performance
would be that sparse rewards need less long term planning. This would
be with the benefit of not accidentally introducing a reward bug due
to shaping, which without knowing the dynamics, cannot be performed
safely in general.



**Time Spent Reviewing:**

4

---

> ### Author Response · Authors · 2021-08-10
> **Response to Reviewer WvH4**
>
> We thank the reviewer for their comments and also for pointing out that the work could have implications on interpretability and verifiability of the agent. These are certainly exciting directions for future work. We address the comments raised by the reviewer below.
>
> **I. Comparison against Off-the-shelf Hierarchical RL:** We agree with the reviewer that a direct comparison against an off-the-shelf hierarchical RL (HRL) approach would be valuable. This, however, is challenging to set up since HRL requires the user to provide reward functions that correspond to the specifications. As motivated by this work and several others, manually designing reward functions for complex specifications is challenging. Instead, we used HRM as one of our baselines. HRM is a hierarchical approach which uses Reward Machines (RMs) to express and decompose the specification [1]. HRM can be viewed as an off-the-shelf hierarchical RL algorithm except that there it uses some tricks to define the low-level rewards and to reuse samples.
>
> As suggested by Reviewer YbXN, in order to further demonstrate the benefits of our compositional algorithm, we will add an empirical comparison to [2] which is a hierarchical RL algorithm that also leverages the abstract structure of the problem. We ran [2]’s algorithm on the motivating example (Figure 1) and for this example DiRL requires ~$2\times10^6$ samples as opposed to ~$5\times10^6$ samples required by [2]. We expect to see similar trends on other benchmarks as well.
>
>
> **II. Path Policies:** The reviewer has raised a keen observation about what happens if an edge policy fails to satisfy an edge. In practice, path policies are executed for a finite horizon. During execution, if an edge policy is unable to satisfy the edge (either violates the constraints or does not reach the goal) the run will be discarded after a finite amount of time and re-run from the beginning. Please note that the probability with which a path policy satisfies the specification (or an edge) gives an estimate of how many runs will be discarded. We will clarify this in the revision.
>
> During training, if no run is found to satisfy the edge, the probability of the edge is assigned to zero. Hence any path that uses it would incur infinite cost. This enables DiRL to identify edges which are not physically possible in the underlying MDP and compute a plan that avoids these edges.
>
> **III.  Alternatives to Reward Shaping for Sparse Rewards:** To use algorithms like HER instead of reward shaping to learn edge policies, which inherently have sparse rewards, is an interesting suggestion. One challenge that an HER-based approach would have to overcome is to incorporate safety constraints. To the best of our knowledge, HER only supports pure reachability tasks. This is certainly a promising direction for future work in RL from specifications.
>
> **IV. Regarding the assumption:** Yes, the assumption we make is reasonable in simulations since we only require the ability to start a simulation from states that were reached using path policies learnt so far. We would like to bring to the reviewers’ attention that this assumption can be removed albeit at a cost. More concretely, starting the MDP from (an already encountered) subgoal region can be simulated by executing the current best path policy to the subgoal region under the MDP’s initial state distribution.
>
> **References:**
> [1]: “Reward Machines: Exploiting Reward Function Structure in Reinforcement Learning”, Rodrigo Toro Icarte, Toryn Q. Klassen, Richard Valenzano, and Sheila A. McIlraith.
> [2]: "Abstract Value Iteration for Hierarchical Reinforcement Learning", Kishor Jothimurugan,  Osbert Bastani, and Rajeev Alur.

---

### Official Review · Reviewer_YbXN · 2021-07-20

**Rating:** 6
**Confidence:** 4

**Summary:**

This paper studies the problem of using logical specifications of a task to derive an abstract graph, come up with a plan, and train low-level policies more effectively. Given a logical specification written in SPECTRL syntax, it is first converted into an abstract MDP. Dijkstra's algorithm is executed on this abstract MDP, assigning probabilities to edges when they are encountered for the first time. A policy is trained for each edge (S_0, S_1) in the abstract MDP where a random state in S_0 is the initial state and a random state in S_1 is the goal state. The success probability of the policy is used to assign the probability of the corresponding edge. The model is evaluated on two rooms environments and a fetch environment from OpenAI gym. The model is shown to outperform previous models.

**Limitations And Societal Impact:**

The paper makes several assumptions. In addition to the ones listed in the paper, it assumes logical specifications can be converted into an abstract mdp deterministically and a set of states that correspond to an abstract state is known.

**Main Review:**

Originality: It is not clear how the proposed DIRL model differs from available abstract MDP literature. Different from previous logical specification literature, this work explores utilizing abstract MDPs for effective planning. The paper mentions that it is different from previous abstract mdp literature since this work focuses on learning from logical specifications. I believe, if the corresponding abstract mdp is given, this would be a planning on abstract mdp problem where there is only a few related work in the paper. Especially, [1] is very relevant to this paper and shares some similarities. Please compare and clarify.

Quality:
There are multiple assumptions in the paper that makes the problem a bit synthetic and not applicable to broader RL problems. It feels like the problem is very simplified as conversion from logical specifications to abstract mdp, planning on the abstract mdp, and training the agent on sub-tasks are all straightforward. I detailed some of these below.

(i) Mapping from sub-regions to abstract states is assumed to be given. This is a crucial assumption where both terminal conditions for each subtask and execution of the high level plan depends on.

(ii) The paper assumes the simulator can be modified. This allows the model to sample states more uniformly, especially since the set of states that correspond to a given abstract state is given. It is not clear if previous work that the paper compares also utilizes this. Please clarify.

(iii) Is the safety constrain really required? There might be different safety requirements in general but this is already implied by the success of the low-level policy (and their edge probabilities) and the example in Figure 1 is more related to task success rather than a safety constrain. It is not clear if this is used in experiments as well. Please clarify.

Clarity: Paper is overall clear but it makes heavy use of notations which at times feel repetitive.

Significance: Results are significant and relevant to the RL community.

[1] "Abstract Value Iteration for Hierarchical Reinforcement Learning", https://arxiv.org/pdf/2010.15638.pdf

**Time Spent Reviewing:**

8

---

> ### Author Response · Authors · 2021-08-10
> **Response to Reviewer YbXN**
>
> We thank the reviewer for the comments and suggestions. We address the reviewer’s comments below.
>
> **I. Comparison to [1] (Abstract Value Iteration):**
> 1. **Expressiveness of the Input:** [1] receives the subgoal regions (abstract states) and an abstract graph as inputs. In contrast, DiRL receives a high-level SpectRL specification as input. The subgoal regions and the abstract graph are inferred automatically from the input specification as part of our solution. The use of SpectRL offers richer expressiveness in sequential tasks than subgoal regions. For instance, SpectRL can express safety constraints which is not possible using subgoal regions alone since safety is expressed over the entire trajectory.
>
> 1. **Algorithm and Sample Complexity:** DiRL is expected to demonstrate improvement in sample complexity. [1]’s algorithm is based on value-iteration while DiRL is based on a simpler Dijkstra’s shortest path algorithm. As a result, [1] learns policies for all edges in the graph while DiRL learns policies only for a subset of the edges that are selected on-the-fly, thereby reducing the sample complexity. Following the reviewer’s comments, we compared the performance of DiRL against [1]. Owing to the limitations of the framework in [1], we made the following adjustments to perform a fair comparison against DiRL:
>    - The sub-goal regions and abstract graphs were provided manually. For a fair comparison, we used the same ones that were generated within DiRL.
>    - A subtle but crucial point here is that [1] requires that the subgoal regions are pairwise disjoint while DiRL does not. Thus, this comparison can only be performed on those benchmarks that generate disjoint subgoal regions in DiRL.
>    - Since the subgoal-region-based framework of [1] does not support safety constraints, we manually encoded the safety constraints in the reward function used to learn edge policies.
>
>     As expected, DiRL demonstrated better sample complexity: On the motivating example (Figure 1) DiRL requires ~$2\times10^6$ sample steps as opposed to ~$5\times10^6$ sample steps required by [1]. We expect to see similar trends on other benchmarks as well. We will include these new experiments on all comparable benchmarks (due to the subtlety of [1] requiring disjoint subgoal regions) in the revised version.
>
> 1. **Assumptions:** [1] assumes the ability to sample states from the subgoal regions. DiRL does not require this assumption, rather, it only requires the ability to detect when a state satisfies a certain predicate and the ability to reset the system to any state that has been observed before. We will include this discussion comparing our work to [1] and other related works on learning from abstract graphs in the revised version.
>
> **II. Real-world Applicability:** We respectfully disagree that our formulation makes the problem synthetic and not broadly applicable. First, we note that a significant amount of recent work in machine learning has focused on learning from specifications, including SpectRL specifications. We do not make any additional assumptions than these approaches (aside from minor assumptions on the simulator), yet we significantly outperform them. Furthermore, the set of benchmarks we consider include both the ones from recent work as well as the ones that are significantly more challenging (in particular, the pick-and-place tasks).
>
> Second, we believe that learning from specifications can capture many important RL tasks that existing algorithms are incapable of solving. In particular, the focus is on long-horizon planning tasks that involve complex sequences of subgoals. In contrast, most existing work in RL has focused on short-horizon tasks such as MuJoCo, which in many ways are more synthetic than the ones we consider. Real-world agents are expected to solve tasks that involve complex sequences of subtasks to achieve a higher level objective. We believe our approach (and more broadly, the line of work on RL from specifications) is a promising way to extend RL to these settings.
>
> **III. Clarification on Assumptions:**
>
> 1. **Mapping from predicates to subgoal regions/abstract states:** We clarify that the mapping from predicates to subgoal regions is NOT manually provided. Instead, DiRL infers this mapping automatically during the construction of the abstract graph from the input specification (Section 3.2 of Main paper and Section A of Supplemental Material). The terminal condition for subtasks is determined by checking whether an MDP state is present in a subgoal region. DiRL automates this as well. Every subgoal region is identified by a set of predicates. Thus, to evaluate whether a state is present in a subgoal region, it is sufficient to evaluate whether the predicate(s) corresponding to the region hold(s) in that state.
>
>     Please note that the ability to evaluate predicates/propositions on MDP states is a commonly used assumption in all prior work on RL from specifications as it is necessary for interpreting specifications on the MDP. Therefore, the only information we require that the user provide is the specification along with the semantics of the predicates in their specification (i.e., an arbitrary function determining whether a concrete state satisfies that predicate). In practice, several alternatives are possible. For instance, rather than provide the predicate semantics, the user can instead provide examples of when the predicate is/isn’t satisfied, in which case we can learn a classifier to serve as the predicate. Such an approach is beyond the scope of our work, but is an interesting direction for future work, and we will add a discussion.
>
> 1. **Assumptions on the simulator:** We do NOT assume that states are sampled more uniformly or that the set of states corresponding to a subgoal region are given. We only assume that trajectories in the MDP can be started from any state that has already been encountered during the execution of a path policy. More precisely, in order to sample a state from a subgoal region, we do the following:
>    * we execute the current best path policy to the subgoal region from the initial states of the MDP,
>    * collect states in the subgoal region encountered in the first step and,
>    * store these states and randomly pick a state among them when we need to simulate the MDP starting at the subgoal region.
>
>     Please note that this simulation model has been used in previous work, for example, [1] and the vine method of TRPO [2]. We would like to bring to the reviewers’ attention that this assumption can be removed albeit at a cost. More concretely, starting the MDP from (an already encountered) subgoal region can be simulated by executing the current best path policy to the subgoal region under the MDP’s initial state distribution.
>
> 1. ***“It assumes logical specifications can be converted into an abstract MDP deterministically and a set of states that correspond to an abstract state is known.”*** We do not assume that logical specifications can be converted to an abstract MDP deterministically. Instead, we prove that learning from SpectRL specifications is reduced to solving abstract reachability on an abstract graph. Similarly, we do not assume that the set of states corresponding to an abstract state is known. As mentioned earlier, the mapping between MDP states and abstract states (subgoal regions) is inferred by the algorithm. Strictly speaking, the only operation required to generate this mapping is the ability to evaluate MDP states against predicates -- which is a standard assumption to interpret specifications. Thus, the set of MDP states corresponding to an abstract state is never computed explicitly.
>
> **IV. Safety Constraints:** Safety constraints and goals are not interchangeable: safety constraints must be satisfied for the duration of the execution of an edge policy, whereas the goal only needs to be achieved by the end of execution. For example, “avoid obstacle” is a safety constraint, and cannot be expressed as a goal, whereas “reach region” is a goal. For a task to be considered successfully completed, it must both achieve its goal AND satisfy the safety constraint; neither by itself is sufficient.
>
> Importantly, we note that in specification-guided RL, environments do not usually enforce safety constraints automatically. For instance, in the Rooms environment, a robot can potentially pass through the red region (the environment doesn’t stop the robot from doing this as there might not be a physical obstacle there). Instead, we must explicitly constrain the policy to avoid the region as part of the specification. The reason is to give the user more control over what safety constraints should be enforced at each subtask. For instance, we could in principle include the possibility that the robot can push a button to deactivate an obstacle, in which case subsequent subtasks would not need to avoid that obstacle region. If safety constraints are enforced by the environment, then we are significantly reducing the flexibility of specification-guided RL.
>
>    Safety constraints are enforced by the “ensuring” clause in SpectRL specifications. We make use of safety constraints in almost all examples on the Rooms Environment. For instance, in the Motivating Example from Figure 1, the safety constraint enforces avoidance of the red regions. A full description of the environments and specifications have been made available in the supplemental material. We will also clarify this in the revision.
>
> **References:**
> [1]: "Abstract Value Iteration for Hierarchical Reinforcement Learning", Kishor Jothimurugan,  Osbert Bastani, and Rajeev Alur.
> [2]: “Trust Region Policy Optimization”, John Schulman, Sergey Levine, Philipp Moritz, Michael I. Jordan and Pieter Abbeel.

---

> > ### Comment · Reviewer_YbXN · 2021-09-02
> > **Thanks for the detailed response**
> >
> > I appreciate this detailed response and new results. It helped me understand your work better and it would also help if you can integrate these into the paper. I am increasing my score to 6.

---

### Decision · Program_Chairs · 2021-09-27

**Decision:**

Accept (Poster)

**Comment:**

This paper presents an approach to learning to solve a task from logical specifications by defining an abstract graph, planning on it, and learning the required low-level skills.
There were several concerns raised by the reviewers that were largely addressed in the discussion, leading to a positive evaluation of the paper.
Specifically, there were questions about the exact contribution, which the authors should make explicit in the final paper. The authors are expected to make the promised changes, including the suggested background and additional baselines, as well as further providing further discussions on the assumptions.